# Defining cellular population dynamics at single-cell resolution during prostate cancer progression

Alexandre A Germanos[1,2], Sonali Arora[1], Ye Zheng[3], Erica T Goddard[4], Ilsa M Coleman[1], Anson T Ku[5], Scott Wilkinson[5], Hanbing Song[6], Nicholas J Brady[7], Robert A Amezquita[3], Michael Zager[8], Annalysa Long[4], Yu Chi Yang[1], Jason H Bielas[4], Raphael Gottardo[3,4], David S Rickman[7], Franklin W Huang[6], Cyrus M Ghajar[1,4], Peter S Nelson[1,9], Adam G Sowalsky[5], Manu Setty[10,11], Andrew C Hsieh[1,9]*

[1]Division of Human Biology, Fred Hutchinson Cancer Center, Seattle, United States; [2]University of Washington Molecular and Cellular Biology Program, Seattle, United States; [3]Division of Vaccine and infectious Diseases, Fred Hutchinson Cancer Center, Seattle, United States; [4]Division of Public Health Sciences, Translational Research Program, Fred Hutchinson Cancer Center, Seattle, United States; [5]Laboratory of Genitourinary Cancer Pathogenesis, Center for Cancer Research, National Cancer Institute, NIH, Bethesda, United States; [6]Division of Hematology/Oncology, Department of Medicine, University of California, San Francisco, San Francisco, United States; [7]Department of Pathology and Laboratory Medicine, Weill Cornell Medicine, New York, United States; [8]Center for Data Visualization, Fred Hutchinson Cancer Center, Seattle, United States; [9]University of Washington Departments of Medicine and Genome Sciences, Seattle, United States; [10]Translational Data Science Integrated Research Center, Fred Hutchinson Cancer Center, Seattle, United States; [11]Division of Basic Sciences, Fred Hutchinson Cancer Center, Seattle, United States

*For correspondence:
ahsieh@fredhutch.org

Competing interest: The authors declare that no competing interests exist.

**Abstract** Advanced prostate malignancies are a leading cause of cancer-related deaths in men, in large part due to our incomplete understanding of cellular drivers of disease progression. We investigate prostate cancer cell dynamics at single-cell resolution from disease onset to the development of androgen independence in an in vivo murine model. We observe an expansion of a castration-resistant intermediate luminal cell type that correlates with treatment resistance and poor prognosis in human patients. Moreover, transformed epithelial cells and associated fibroblasts create a microenvironment conducive to pro-tumorigenic immune infiltration, which is partially androgen responsive. Androgen-independent prostate cancer leads to significant diversification of intermediate luminal cell populations characterized by a range of androgen signaling activity, which is inversely correlated with proliferation and mRNA translation. Accordingly, distinct epithelial populations are exquisitely sensitive to translation inhibition, which leads to epithelial cell death, loss of pro-tumorigenic signaling, and decreased tumor heterogeneity. Our findings reveal a complex tumor environment largely dominated by castration-resistant luminal cells and immunosuppressive infiltrates.

## Editor's evaluation

Prostate cancer cellular heterogeneity is a major problem for disease progression and treatment resistance. This body of work addresses the cellular identity and populations that make up prostate

cancer using single-cell sequencing technology, state-of-the-art mouse models and connections to human prostate cancer. The cellular identities, associated signaling networks, and immune complexes accompanying the heterogeneity of the prostate are identified in this work and a resource is provided for scientists in the field.

## Introduction

Prostate cancer is the most diagnosed malignancy and second leading cause of cancer-related death in men in the United States (*Siegel et al., 2021*). This is largely due to the development of treatment resistant disease termed castration resistant prostate cancer (CRPC). Historically, CRPC has been considered a singular disease entity. However, this viewpoint has significantly evolved with the advent of next generation sequencing and the characterization of many distinct etiologies (*Watson et al., 2015*; *Bluemn et al., 2017*; *Labrecque et al., 2019*). Intratumoral heterogeneity is also common in prostate cancer, as several studies have highlighted both the complex cellular architecture of the prostate and multiple potential cell-of-origin models (*Goldstein et al., 2010*; *Shen and Abate-Shen, 2010*). For instance, while prostate epithelial cells canonically present as either basal cells or highly differentiated luminal cells, a distinct, more stemlike luminal population (*Psca+/Krt4+/Tacstd2+*) that primarily resides in the proximal prostate has previously been described (*Xin et al., 2005*; *Wang et al., 2006*; *Korsten et al., 2009*; *Kwon et al., 2016*; *Sackmann Sala et al., 2017*; *McAuley et al., 2019*). While the nomenclature for this cell type has varied, most studies have observed a similar set of biomarkers and characteristics, including increased stemlike and inflammatory/immunogenic signatures (*Liu et al., 2016*). These cells also appear to be castration-resistant; indeed, some studies show canonical prostatic luminal cells may undergo phenotype switching towards a more stemlike *Krt4+* cell state in castrate conditions, reverting and helping repopulate the prostate during castration and regeneration cycles (*Karthaus et al., 2020*; *Mevel et al., 2020*; *Guo et al., 2020*; *Kwon et al., 2021*).

This novel cell type was further verified using lineage tracing and single-cell technologies (*Liu et al., 2016*; *Crowley et al., 2020*; *Joseph et al., 2020*; *Karthaus et al., 2020*; *Kwon et al., 2020*; *Mevel et al., 2020*; *Guo et al., 2020*). *Krt4+* cells are also found more rarely in the distal prostate, where they localize to invagination tips (*Guo et al., 2020*). These cell types remain incompletely characterized, including in their origin; some studies point to proximal luminal cells originating from the urethra, and therefore not being prostatic in nature (*Joseph et al., 2020*). Others recognize the presence of urethral cells but draw a distinction between them and prostatic *Krt4+* proximal cells (*Crowley et al., 2020*).

There is evidence that *Krt4 +* cells may contribute to tumor initiation in the prostate (*Xin et al., 2005*; *Wang et al., 2006*; *Korsten et al., 2009*; *Kwon et al., 2016*; *Sackmann Sala et al., 2017*; *Guo et al., 2020*). For instance, several studies find these cells enriched in precancerous lesions upon genetic perturbations in murine models (*Wang et al., 2006*; *Korsten et al., 2009*). Additionally, proximal cells appear less able to contribute to tumor initiation than those residing in distal prostate lobes (*Korsten et al., 2009*; *Guo et al., 2020*). These findings, combined with studies characterizing the role of *Krt4+* cells in castration and regeneration, suggest they may contribute to treatment resistance in advanced cancer. However, this has yet to be established in CRPC models.

While tumor heterogeneity contributes to treatment resistance, another source of poor clinical outcomes can be found in the consistent lack of response of CRPCs to immunotherapy. Prostate cancer has generally been described as 'immune-cold' due to the presence of multiple immunosuppressive cell types (*Stultz and Fong, 2021*). For example, tumor infiltration by myeloid-derived suppressor cells (MDSCs) has been implicated as an immunosuppressive phenotype (*Garcia et al., 2014*; *Lopez-Bujanda et al., 2021*). Metastatic CRPC also responds poorly to immune checkpoint inhibition and other immunotherapies (*Cham et al., 2020*). This has been confirmed clinically with autologous active cellular immunotherapy which only demonstrated a small (~4 month) improvement in survival in patients (*Kantoff et al., 2010*). Therefore, characterizing the immune microenvironment in advanced prostate cancer may be crucial to better understand this disease and inform potential therapeutic vulnerabilities or combinatorial strategies.

Here, we have created an atlas of prostate cellular composition and phenotypic evolution through tumor initiation, progression, and hormone independence using a *Pten* loss murine model as an archetype. We observe a dramatic expansion of an intermediate luminal cell type in cancer and

link this epithelial population to treatment resistance in multiple human cohorts. We also characterize increased pro-tumorigenic immune cell recruitment and define cell-cell signaling patterns that contribute to lymphoid and myeloid cell expansion. In addition, using a tissue-specific transgenic model, we demonstrate that cell type specific protein synthesis is essential for the maintenance of tumor heterogeneity in both basal and intermediate cells in the context of AR-low prostate cancer. Finally, we make our data available for further study through an interactive portal, with the aim of providing a broad resource for the cancer research community. Together, our findings highlight the cell type-specific and patient relevant features of prostate cancer progression and demonstrate the utility of single-cell technologies to uncover novel cell-specific paradigms of treatment resistance.

## Results

### Characterization of WT and Pten$^{fl/fl}$ ventral prostates at single cell resolution

To determine the cellular architecture of prostate cancer initiation, we conducted single-cell RNA sequencing (scRNAseq) of the ventral prostates of 6-month-old wild-type (WT) and *PB-Cre4;Pten$^{fl/fl}$;ROSA26-rtTA-IRES-eGFP* (herein referred to as *Pten$^{fl/fl}$*, see Methods) prostate cancer mice (*Figure 1A*). We chose the ventral lobe because it has been well-characterized throughout prostate cancer progression at a deep molecular level and for the abundance of cellular populations (*Hsieh et al., 2012*; *Hsieh et al., 2015*; *Liu et al., 2019*). In the *Pten$^{fl/fl}$* model, exon 5 of the tumor suppressor *Pten* is deleted within basal and luminal epithelial cells of the prostate and mice uniformly develop prostate cancer (*Wang et al., 2003*). PTEN is a negative regulator of the oncogenic PI3K-AKT-mTOR signaling pathway, which is deregulated in nearly all advanced prostate cancer patients (*Taylor et al., 2010*).

Quality control and filtering (read count thresholds and excluding cells with high mitochondrial content) resulted in transcriptomes from 24,079 total cells across five samples (*Supplementary file 1A-C*). Using SingleR, we identified epithelial, stromal, and immune cell types (*Aran et al., 2019*; *Figure 1—figure supplement 1A*). To confirm that the epithelial cells in the *Pten$^{fl/fl}$* model underwent Cre-mediated recombination, we measured the frequency of *rtTA-eGFP* fusion mRNA in WT and *Pten$^{fl/fl}$* mice. The *rtTA-eGFP* transcript was present in 14.5% of *Pten$^{fl/fl}$* epithelial cells (*Figure 1—figure supplement 1B*). Given that on average ~1200 genes/cell were detected and ~24,000 genes were identified overall, we expect the average transcript to be found in ~5% of cells in the dataset. Therefore, our finding suggests widespread recombination, and therefore loss of *Pten*, within the epithelial compartment. Importantly, recombination was not observed in WT mice or non-epithelial cell types (*Figure 1—figure supplement 1B*).

We further defined the epithelial population in WT mice via signatures and biomarkers obtained from the Strand (Basal, Urethral, and VP), Sawyers (Basal, L1, and L2), and Xin (Sca-1+) groups (*Joseph et al., 2020*; *Karthaus et al., 2020*; *Liu et al., 2016*). We found canonical basal (*Krt5+/Sox4+*) and luminal (*Nkx3-1+/Sbp+*) cells, as well as two distinct clusters expressing luminal progenitor markers (*Psca+/Tactsd2+/Krt4+*) (*Figure 1—figure supplement 1C–D*). These clusters were mainly distinguished by different basal, L2, and luminal (VP/L1) signature scores. In addition, one cluster clearly expressed previously identified urethral markers *Aqp3* and *Ly6d*, and was negative for other progenitor markers (*Ppp1r1b, Clu*; *Figure 1—figure supplement 1C*; *Crowley et al., 2020*). Accordingly, we designated this cluster as urethral. The other *Krt4+* cluster likely represents progenitor prostatic cells, either from the proximal or distal prostate (*Crowley et al., 2020*; *Guo et al., 2020*). Therefore, we refer to this cluster as luminal progenitor hereafter (*Figure 1—figure supplement 1D*).

We then examined epithelial cell subtypes in *Pten$^{fl/fl}$* mice. We identified basal, urethral, and canonical luminal cells, as well as a large cluster of cells similar to WT luminal progenitor cells, characterized by expression of *Krt4/Tacstd2/Ppp1r1b* but not *Psca* (*Figure 1—figure supplement 1E–F*). Clustering urethral cells with luminal progenitor and this new *Pten$^{fl/fl}$* cluster, we find that the urethral cells cluster apart from the other cell types and express significantly lower levels of prostate luminal or pan-epithelial markers (*Krt8, Krt18, Epcam*) (*Figure 1—figure supplement 1G–H*). This supports the hypothesis that we have identified both urethral and prostate cells, and likely indicates a prostatic origin for the *Pten$^{fl/fl}$* cluster. We designated *Krt4+/Tacstd2+/Ppp1r1b+* cells in *Pten$^{fl/fl}$* mice as 'intermediate' cells (*Figure 1—figure supplement 1E*).

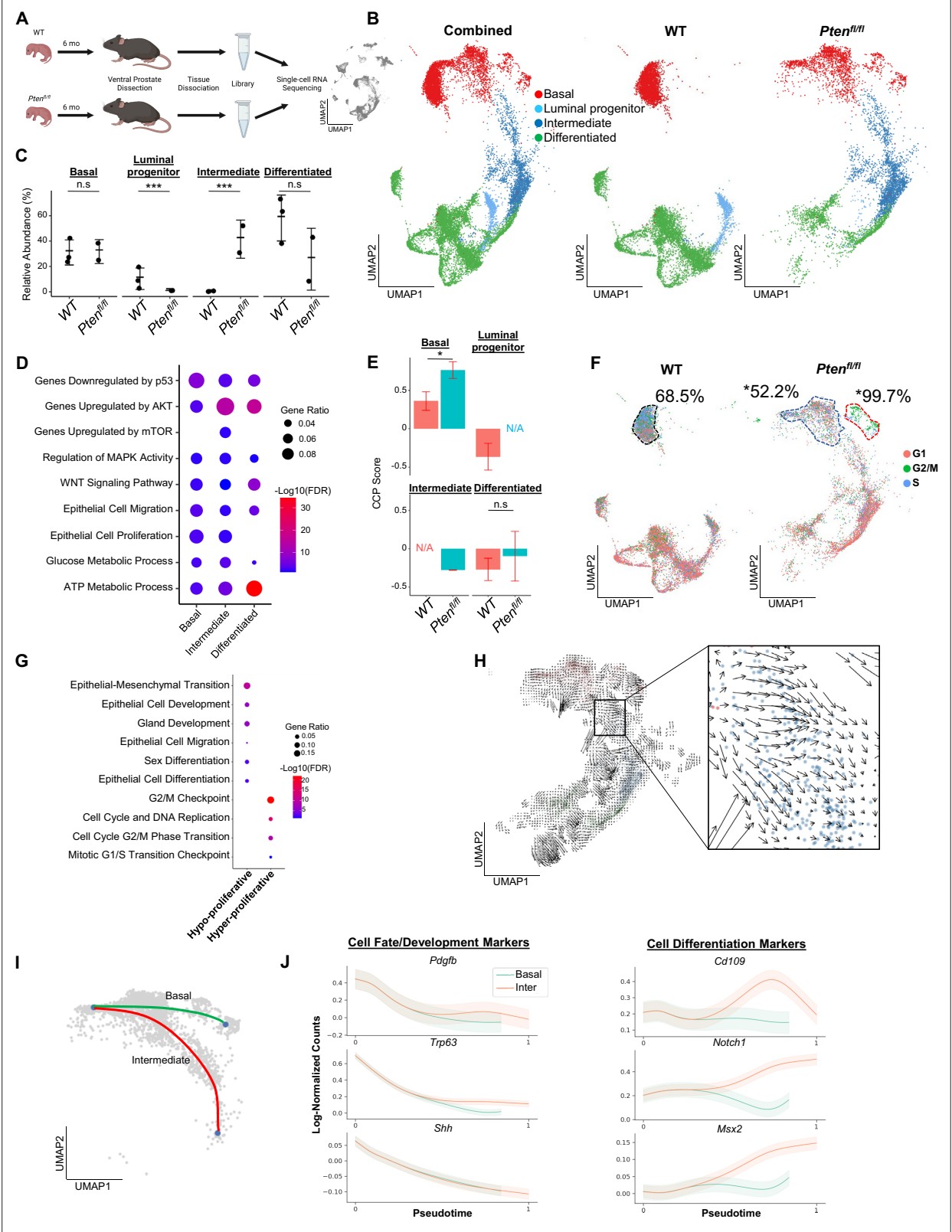

**Figure 1.** Proliferative split in basal cancer cells enables expansion of intermediate cells. (**A**) Simplified schematic of single-cell RNA sequencing of WT and *Pten^fl/fl* ventral prostates. (**B**) UMAP of WT and *Pten^fl/fl* epithelial cells. Left, both conditions superposed; middle, WT only; right, *Pten^fl/fl* only. Epithelial cell types are demarcated by color (red = basal, light blue = luminal progenitor, dark blue = intermediate, green = differentiated). (**C**) Relative abundance of epithelial cells in WT (n=3) and *Pten^fl/fl* (n=2) mice. Y-axis shows the % composition of each sample by cell type (***p<0.001, negative

*Figure 1 continued*

binomial test). Data presented as +/-SD. (**D**) Top GSEA results enriched in *Pten*<sup>fl/fl</sup> compared to WT for each epithelial subtype. Intermediate cells in *Pten*<sup>fl/fl</sup> were compared to luminal progenitor cells in WT. All pathways are enriched with FDR <0.05. (**E**) Proliferation signature (CCP) composite score in epithelial cells, clustered by condition (Data presented as +/-SD, *p<0.05, n.s.=not significant, permutation test). N/A indicates missing data due to no cells being present in the condition. WT n=3, *Pten*<sup>fl/fl</sup> n=2. (**F**) UMAP visualization of cell cycle phase assignment per cell, showing % cells in non-G1 (S or G2/M) (black border = WT basal cells, blue border = hypo-proliferative basal cells in *Pten*<sup>fl/fl</sup>, and red border = hyper-proliferative basal cells in *Pten*<sup>fl/fl</sup>. *p<0.05, chi-square test). (**G**) GSEA between hyper- and hypo-proliferative basal clusters in *Pten*<sup>fl/fl</sup>. All pathways are enriched with FDR <0.05. (**H**) RNA velocity analysis of *Pten*<sup>fl/fl</sup> epithelial cells; highlighted section shows intersection of basal and intermediate cells. (**I**) Pseudotime trajectories drawn by Palantir through the basal and intermediate compartments, with hypo-proliferating basal cells as the designated start point. (**J**) Expression of important cell fate and differentiation regulators along basal-intermediate trajectory.

The online version of this article includes the following figure supplement(s) for figure 1:

**Figure supplement 1.** Epithelial cells contain published subtypes and urethral cells.

**Figure supplement 2.** Basal proliferation is subset-specific and intermediate cells express luminal markers.

## *Pten* loss generates differentially proliferating populations and a dominance of intermediate luminal cell states from multiple cellular origins

Next, we sought to determine how the epithelial compartment is remodeled in the context of tumor initiation. We excluded the urethral cluster because it is not prostatic in origin and therefore is less likely to significantly impact cellular dynamics in prostate cancer. We observed a dramatic increase in the proportion of intermediate cells in *Pten*<sup>fl/fl</sup> mice compared to luminal progenitor abundance in the WT setting (*Figure 1B–C*). To investigate potential mechanisms of this expansion of the intermediate population, we identified differentially expressed genes (DEG) and performed gene set enrichment analysis (GSEA) on each epithelial subtype, comparing the WT and *Pten*<sup>fl/fl</sup> epithelial compartments (*Pten*<sup>fl/fl</sup> basal:WT basal, *Pten*<sup>fl/fl</sup> intermediate:WT luminal progenitor, *Pten*<sup>fl/fl</sup> luminal:WT luminal) (*Supplementary file 1D*). We noted enrichment of oncogene and tumor suppressor pathways in the *Pten*<sup>fl/fl</sup> mice, including upregulation of AKT and mTOR, which are expected in this model. We also observed increased *Wnt* signaling and metabolic processes across all epithelial cell types. Interestingly, both epithelial cell migration and proliferation were enriched in most epithelial cells (*Figure 1D*). These findings suggest that intermediate cell expansion could be mediated by increased proliferation of luminal progenitors, or by increased transdifferentiation of other cell types, which has been widely characterized in WT settings (*Karthaus et al., 2020*; *Mevel et al., 2020*; *Kwon et al., 2021*).

To evaluate the possibility of increased proliferation of different epithelial cell types, we used the Cell Cycle Progression (CCP) score, a proliferation gene signature developed and validated in human prostate cancer patients (*Cuzick et al., 2011*), and generated a composite score (see Materials and methods) in our dataset. Surprisingly, while there was a significant increase in CCP score in basal cells in *Pten*<sup>fl/fl</sup> mice compared to WT, we observed no significant change in intermediate cells compared to luminal progenitor cells (*Figure 1E*). To further characterize epithelial proliferation, we performed cell cycle scoring, assigning one of three phases (G1, G2/M, or S) to each cell in the dataset, and found a striking split in the basal compartment of *Pten*<sup>fl/fl</sup> mice, but not WT mice (*Supplementary file 1E*). In *Pten*<sup>fl/fl</sup> mouse prostates, 18.6% of basal cells were hyper-proliferative, with 99.7% of these cells in a proliferative phase (G2/M or S phases). The rest of the basal compartment only had 52.2% of cells in a G2/M or S phase, lower than WT basal cells (68.5%) (*Figure 1F*, *Figure 1—figure supplement 2A*). We also verified that the increase in basal cell CCP score was specifically due to this hyper-proliferative cluster (*Figure 1—figure supplement 2B*). One possibility for this differential increase in basal cell proliferation is higher Cre-mediated recombination efficiency in some clusters over others. We analyzed transgene abundance in our basal subclusters and found that in the *Pten*<sup>fl/fl</sup> mouse, 13.6% (12.0%–15.7%) of hypo-proliferating and 19.6% (19.1%–20.3%) of hyper-proliferating basal cells express the *rtTA-eGFP* transgene. This <1.5-fold difference does not account for the >twofold increase in cycling cells, or the ~threefold increase in CCP score observed between the subclusters. As such, we conclude that *Pten* loss in the murine prostate promotes a dual phenotype in basal cells, with most cells displaying decreased proliferation while a subset becomes hyper-proliferative.

We further characterized the basal subclusters in *Pten*<sup>fl/fl</sup> mice by performing DEG analysis followed by GSEA (*Supplementary file 1F*). As expected, we observed several cell-cycle-related pathways

enriched in the hyper-proliferative cluster. However, the hypo-proliferative basal cluster was enriched for several migration, development, and differentiation signatures (*Figure 1G*). Transdifferentiation of basal cells to luminal cells in the context of malignant transformation has previously been reported in prostate cancer (*Goldstein et al., 2010*; *Choi et al., 2012*; *Lu et al., 2013*; *Wang et al., 2013*). As such, we hypothesized that the hypo-proliferative basal subcluster might transition into intermediate luminal cells, thus providing one potential mechanism for the expansion of this cell type in the absence of increased proliferation. To verify this, we generated pseudotime trajectories using Monocle3 (*Cao et al., 2019*), which confirmed a direct path from hypo-proliferative basal cells to intermediate luminal cells (*Figure 1—figure supplement 2C*). We also performed RNA velocity, a technique to visualize differentiation dynamics on a per cell basis (*La Manno et al., 2018*). This analysis revealed significant movement from hypo-proliferative, but not hyper-proliferative basal cells to intermediate cells (*Figure 1H*). Finally, we used a pseudotime algorithm (*Setty et al., 2019*) to identify two potential trajectories starting from the hypo-proliferative basal cells and ending either in the hyper-proliferative basal cluster or in the intermediate luminal cell compartment (*Figure 1I*). We generated clusters of genes with similar expression patterns across these pseudotime axes (*Figure 1—figure supplement 2D*) and performed GSEA on the clusters. We found that several genes involved in cell fate decisions including *Pdgfb*, *Trp63*, and *Shh* are highly expressed early but rapidly decrease over the trajectories. Conversely, genes implicated in differentiation such as *Notch1* increased over the basal-intermediate path, but not the basal-basal path (*Figure 1J*). This suggests that hypo-proliferating basal cells strongly express development markers associated with cell fate choice, but these genes are not expressed in hyper-proliferative basal cells or intermediate cells. Moreover, intermediate cells express higher levels of differentiation-associated genes than either basal subcluster. Together, these findings support the idea that hypo-proliferative basal cells transdifferentiate into intermediate luminal cells during tumorigenesis, while the basal compartment may be replenished by a hyper-proliferative, non-differentiating population.

Multiple studies have suggested that prostate luminal cells can phenotype switch to intermediate-like states under conditions such as castration (*Karthaus et al., 2020*; *Guo et al., 2020*). To verify whether luminal populations may contribute to the intermediate cell state, we plotted the expression of genes canonically expressed in prostate luminal cells and found that while luminal cells did express these genes most strongly, there was overlap with a population of intermediate cells immediately adjacent to the luminal cells in *Pten*$^{fl/fl}$ mice (*Figure 1—figure supplement 2E*). This suggests that luminal cells may phenotype switch into an intermediate state and retain some identity markers in the context of *Pten* loss. It is also possible that luminal progenitor cells contribute to the intermediate population, as *Krt4+* cells within the proximal and distal prostate have been described as tumor initiating cells after *Pten* loss (*Guo et al., 2020*). Overall, our findings support multiple cells of origin for *Krt4+* intermediate cells in *Pten*$^{fl/fl}$ mice, including phenotype switching of basal and luminal cells, as well as potential expansion of luminal progenitor cells. In vivo lineage tracing studies using highly precise Cre-drivers will likely be necessary to fully disentangle the cellular origins of intermediate cell states in the *Pten*$^{fl/fl}$ prostate cancer model.

## Immune infiltration increases in *Pten*$^{fl/fl}$ mice and is pro-tumorigenic

Our GSEA of all three epithelial cell types showed an enrichment for immune-related pathways in *Pten*$^{fl/fl}$ mice, suggesting increased immunogenic signaling relative to WT mice (*Figure 2A*). To determine the impact of epithelial *Pten* loss on immune cells, we calculated the relative abundance of immune cells in WT and *Pten*$^{fl/fl}$ mice. We found that T cells, macrophages, neutrophils, and dendritic cells were all significantly expanded in the *Pten*$^{fl/fl}$ mouse; in fact, only B cells were not expanded compared to WT (*Figure 2—figure supplement 1A–B*). Immune infiltration has previously been characterized as immunosuppressive in PTEN null prostate tumors (*Garcia et al., 2014*). Therefore, we expected these expanding immune populations to be immunosuppressive and therefore pro-tumorigenic. To test this hypothesis, we assigned activation states or subtypes to immune cells based on known biomarkers (*Figure 2B*, *Figure 2—figure supplement 1C–E*). First, we characterized the neutrophil population as myeloid-derived suppressor cells (MDSCs) based on published biomarkers (*Alshetaiwi et al., 2020*; *Figure 2—figure supplement 1C*). We also found three macrophage cell states: tumor-associated macrophages (TAMs), M2-activated macrophages, and M1-activated macrophages. Interestingly, the M1 cells expressed the AR-dependent markers *Sbp* and *Defb50* and as

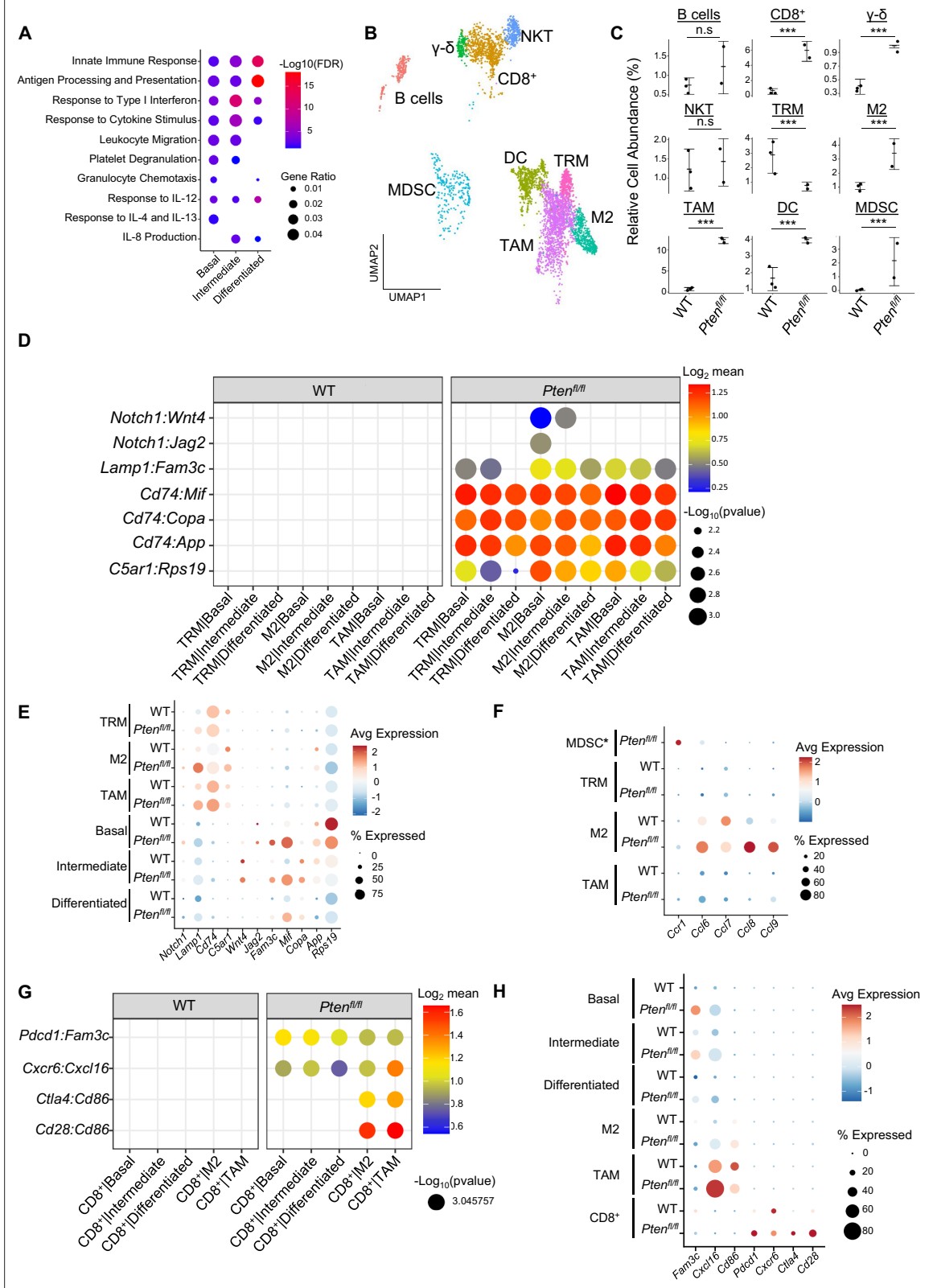

**Figure 2.** Immune recruitment in *Pten*<sup>fl/fl</sup> prostates is mediated by both epithelial and immune cell signaling. (**A**) Top immune-related GSEA results enriched in *Pten*<sup>fl/fl</sup> compared to WT mice for each epithelial subtype. All pathways are enriched with FDR <0.05. (**B**) UMAP visualization of immune cells labeled by cell subtype or state. (**C**) Relative abundance of immune cells in WT (n=3) and *Pten*<sup>fl/fl</sup> (n=2) mice. Y-axis shows the % composition of each sample by cell type (Data presented as +/-SD, ***p<0.001, n.s.=not significant, negative binomial test). (**D**) Dot plot of signaling interactions between

*Figure 2 continued on next page*

*Figure 2 continued*

macrophages and epithelial cells. Y-axis, ligand-receptor pairs from CellphoneDB database. X-axis, cell-cell pairings. Interactions are directional: the first gene in a pair is expressed in the first cell in the cell-cell interaction. (**E**) Dot plot of epithelial ligand and macrophage receptor gene expression in WT and *Pten*$^{fl/fl}$ mice. (**F**) Dot plot of *Ccr1* and *Ccr1* ligand expression in MDSCs and macrophages in WT and *Pten*$^{fl/fl}$ ventral prostates. MDSCs are only present in *Pten*$^{fl/fl}$ and therefore do not have a WT row (denoted by asterisk). (**G**) Plot of signaling interactions between CD8 +T cells and epithelial cells and macrophages. (**H**) Dot plot of CD8$^+$ T cell receptors and epithelial and macrophage ligand gene expression in WT and *Pten*$^{fl/fl}$ ventral prostates.

The online version of this article includes the following figure supplement(s) for figure 2:

**Figure supplement 1.** Immune cells contain pro-tumorigenic subtypes and macrophages are recruited by fibroblast signaling.

a result, we termed them tissue-resident macrophages (TRM, *Figure 2—figure supplement 1D*). Finally, we characterized three T cell subtypes: CD8$^+$ T cells, gamma-delta T cells, and natural killer T (NKT) cells. CD8$^+$ and gamma-delta T cells expressed the markers of exhaustion and immunosuppression *Pdcd1* and *Ctla4*, suggesting their cytotoxic activity may be dampened in the context of *Pten* loss (*Figure 2—figure supplement 1E*). All these cell types except for NKT cells and TRMs are expanded in *Pten*$^{fl/fl}$ mice, implying a potential role for these immune cells in establishing or maintaining a pro-tumorigenic prostate tumor microenvironment (*Figure 2C*).

M2 macrophages, TAMs, and MDSCs are generally pro-tumorigenic (*Zaynagetdinov et al., 2011*; *Chanmee et al., 2014*; *Garcia et al., 2014*). Together with exhausted CD8$^+$ T cells, this suggests a broadly pro-tumorigenic immune environment. We hypothesized that cell-cell signaling originating from tumor epithelial cells may play an important role in recruiting immune cells to the prostate. To probe cell-cell interactions in our system, we used a ligand-receptor database and interaction algorithm that classifies the strength of specific ligand-receptor interactions between cell groups (Efremova et a., 2020). We found key interactions between epithelial subtypes and M2 macrophages and TAMs, such as interactions targeting the *Cd74* receptor, that point to active recruitment (*Figure 2D*, *Supplementary file 2*). Ligands in the epithelial basal, intermediate, and differentiated cell populations were more highly expressed in *Pten*$^{fl/fl}$ mice than in WT mice, which corresponds to the increased macrophage abundance in *Pten*$^{fl/fl}$ mice (*Figure 2E*).

Additionally, we observed increased *Ccl2/7/11-Ccr2* interactions between fibroblasts and M2/TAMs upon *Pten* loss, suggesting that fibroblasts in prostate cancer may also play an active role in macrophage recruitment (*Figure 2—figure supplement 1F*). *Ccl2/7/11* are all significantly upregulated in *Pten*$^{fl/fl}$ fibroblasts (*Figure 2—figure supplement 1G*). Interestingly, *Ccr2* is expressed highly in M2 and TAMs in cancer, but not in TRMs (*Figure 2—figure supplement 1H*). The lack of fibroblast signaling to TRMs provides a possible cellular explanation for the lower abundance of this macrophage subtype in *Pten*$^{fl/fl}$ mice compared to WT. Another possibility is that TRMs polarize and transition into M2/TAM macrophages, potentially in response to epithelial signaling. Overall, these findings highlight a role for epithelial cells and fibroblasts in recruiting pro-tumorigenic macrophages to the prostate in the context of *Pten* loss.

While we observed significant epithelial to macrophage signaling, the top interactions between epithelial cells and MDSCs were characterized by receptors expressed in epithelial cells and ligands expressed in MDSCs. As such, it is unlikely that epithelial cells are responsible for the expansion of MDSCs in the prostate (*Figure 2—figure supplement 1I*). Instead, we noted increased expression of the *Ccr1* receptor in MDSCs, and a concomitant increase in *Ccl6/7/8/9* expression in M2 macrophages. These genes are known *Ccr1* ligands (*Korbecki et al., 2020*), which suggests that MDSCs are recruited by M2-activated macrophages (*Figure 2F*). Interestingly, *Ccl6/7/8/9* are not expressed in TRMs or TAMs, highlighting the specificity of M2-MDSC signaling.

Finally, we examined interactions between epithelial cells and T cells. We found both attractive (*Cxcr6-Cxcl16*) and pro-exhaustion (*Pdcd1-Fam3c*) interactions between CD8$^+$ T cells and epithelial cells. We found these same interactions between macrophages and CD8$^+$ cells, as well as *Cd86-Ctla4/Cd28* interactions (*Figure 2G*). *Ctla4* and *Cd28* belong to the same family of T cell co-stimulator receptors. *Ctla4* has a suppressive effect while *Cd28* is activating, and both receptors are targeted by the *Cd86* ligand (*Gmünder and Lesslauer, 1984*; *Salomon and Bluestone, 2001*; *Boesteanu and Katsikis, 2009*). These findings imply a synergistic signaling pattern with both tumor epithelia and macrophages recruiting and mitigating the cytotoxicity of T cells. We noted that TAMs exhibited particularly robust expression of *Cxcl16* and *Cd86*, suggesting that similar to M2-mediated MDSC

recruitment, TAMs may specifically be responsible for modulating T-cell abundance and function (*Figure 2H*).

Overall, our data suggests that pro-tumorigenic macrophages are recruited by epithelial and fibroblast signaling in *Pten* null tumors. These macrophages then assist tumor signaling in remodeling the immune environment, including recruiting and exhausting cytotoxic CD8+ T cells and attracting pro-tumorigenic MDSCs. Given ligand expression data, M2 macrophages may be mainly responsible for MDSC recruitment and TAMs for CD8+ T cell recruitment. These findings reveal a complex signaling system with multiple coordinated sources of ligand expression working in tandem to build a microenvironment favorable to tumor escape from immunological suppression.

## Castration-induced intermediate cell heterogeneity drives resistance to androgen deprivation

Having examined epithelial and immune populations in prostate cancer initiation, we sought to determine how these cells reorganize over the course of castration resistance. It has been shown that castration of *Pten*<sup>fl/fl</sup> mice leads to the emergence of AR-low tumors (*Liu et al., 2019*). As such, we conducted scRNAseq in 6 months old *Pten*<sup>fl/fl</sup> mice with and without castration at 4 months of age and evaluated the changes that occurred in the epithelial compartment (*Figure 3A*). Castration caused the intermediate luminal cell population to expand while the differentiated luminal cells disappeared entirely (*Figure 3B–C*). We hypothesized that androgen deprivation may differentially affect epithelial subtypes. To investigate this, we generated an AR activity score using a 20-gene signature (*Hieronymus et al., 2006*) and found that WT luminal progenitor cells had high AR activity, but *Pten*<sup>fl/fl</sup> intermediate cells exhibited very low AR activity in both intact and castrated *Pten*-null conditions. On the contrary, differentiated luminal cells retained high AR activity in *Pten*<sup>fl/fl</sup> mice (*Figure 3D*). This suggests that loss of *Pten* in intermediate luminal cells decreases their reliance on AR signaling and induces expansion of this compartment upon castration. While it is possible that castration is lethal to differentiated luminal cells, previous studies have shown that they are able to phenotype switch into an intermediate state in AR-low conditions (*Karthaus et al., 2020*). Therefore, it is likely that this increase in intermediate cell abundance in *Pten*<sup>fl/fl</sup> castrate mice is due to lineage plasticity in luminal cells. Notably, there were no significant changes in proliferation between the intact and castrate conditions in basal or intermediate cells (*Figure 3—figure supplement 1A*).

Given the increase in intermediate cells associated with castration resistant tumor growth, we next asked how castration modulates phenotypic diversity in this population. We isolated and re-clustered intermediate cells from *Pten*<sup>fl/fl</sup> intact and castrated mice and found six distinct clusters (*Figure 3E*). The majority of intact *Pten*<sup>fl/fl</sup> intermediate cells (62.8%) congregated in a single cluster (cluster 3), while castrated *Pten*<sup>fl/fl</sup> intermediate cells were widely distributed over four unique groups (clusters 0, 1, 2, and 4) (*Figure 3F*). DEG analysis showed high expression of AR-dependent genes *Sbp*, *Defb50*, and *Spink1* in cluster 3, suggesting this cell population may have high AR signaling activity relative to other intermediate cells (*Figure 3—figure supplement 1B*, *Supplementary file 3A-F*). This cluster represents cells close to differentiated luminal cells on the epithelial UMAP (*Figure 3B*) and may indicate a luminal origin that could explain retaining high AR activity. Similarly, we observed high expression of the basal cell markers *Krt5* and *Krt15* in cluster 1 (*Figure 3—figure supplement 1B*; this cluster corresponds to the intermediate cells proximal to basal cells on the epithelial UMAP (*Figure 3B*)). This supports our hypothesis that some intermediate cells are derived from basal transdifferentiation in the context of cancer, while others may have luminal origins. We also noted multiple ribosomal genes upregulated in clusters 1 and 4, suggesting increased translational signatures in these clusters (*Figure 3—figure supplement 1B*). GSEA confirmed enrichment of multiple translation pathways in clusters 1 and 4 (*Figure 3—figure supplement 1C*). We also investigated whether castration leads to increased translational signatures in basal cells. We found that several genes encoding ribosomal proteins were overexpressed in castrated basal cells compared to intact *Pten*<sup>fl/fl</sup> mice (*Figure 3—figure supplement 1D*). Together, these findings demonstrate that castration promotes increased heterogeneity of the intermediate cell populations and potentially diversifies cell-type-specific translation dependence in both basal and intermediate cells in the context of *Pten* loss.

To further characterize the functional differences in castration resistant intermediate cells, we generated AR activity, proliferation, and translation scores for each intermediate cell cluster. We found a gradient of residual AR signaling activity, with cluster 3 having the highest score (*Figure 3G*).

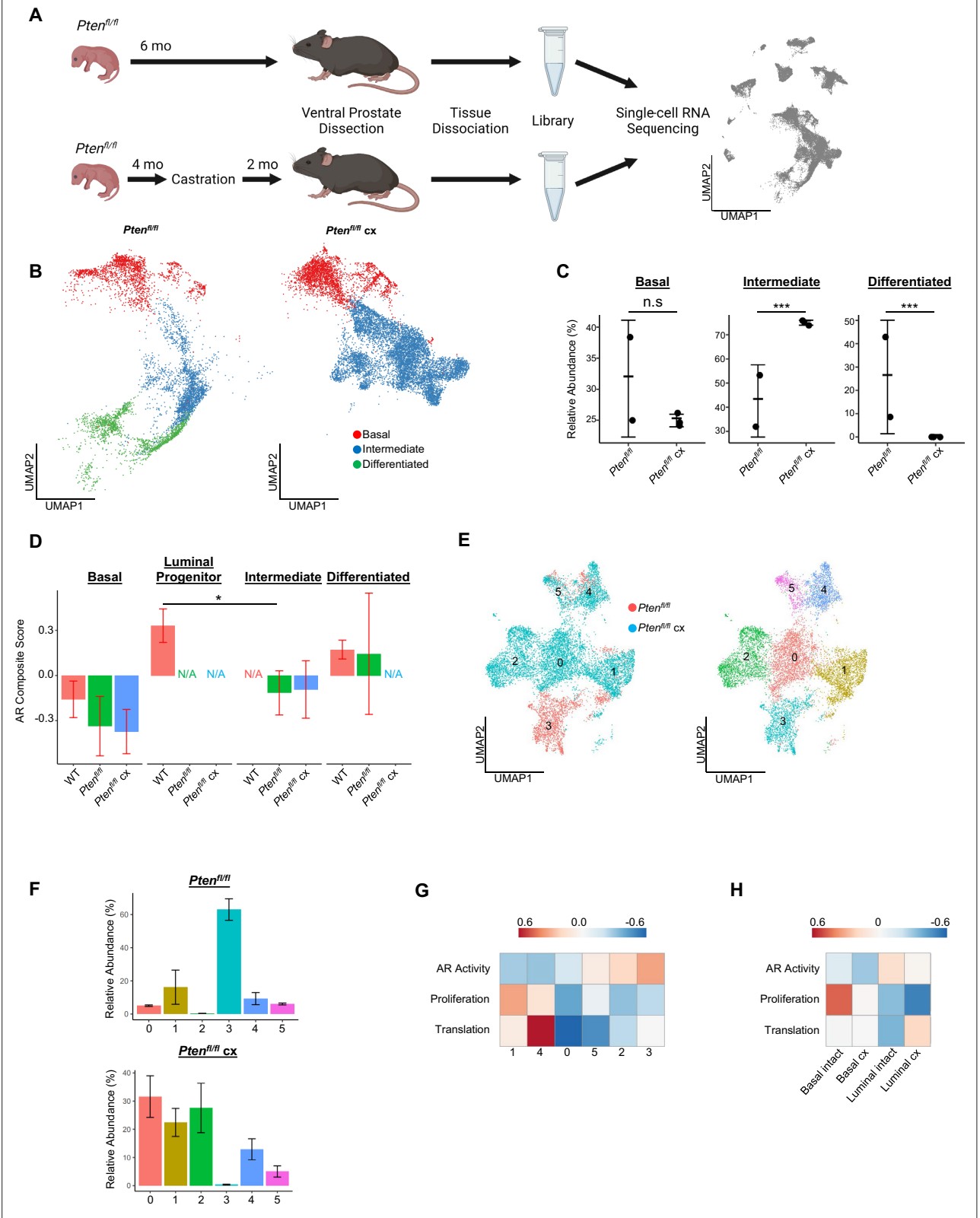

**Figure 3.** Intermediate cells are primed for survival and diversification in the context of castration. (**A**) Simplified schematic of setup for single-cell sequencing of *Pten*^fl/fl^ intact and *Pten*^fl/fl^ castrated (cx) ventral prostates. (**B**) Split UMAP visualizations of *Pten*^fl/fl^ and *Pten*^fl/fl^ cx epithelial cells. (**C**) Relative abundance of epithelial cells in *Pten*^fl/fl^ intact (n=2) and cx (n=3) prostates. Y-axis shows the % composition of each sample by cell type (Data presented as +/-SD, ***p<0.001, n.s.=not significant, negative binomial test). (**D**) Androgen Receptor (AR) gene signature composite score in epithelial cells,

*Figure 3 continued on next page*

*Figure 3 continued*

clustered by condition (Data presented as +/-SD, *p<0.05, permutation test). N/A indicates missing data due to no cells being present in the condition. WT n=3, *Pten^{fl/fl}* intact n=2, *Pten^{fl/fl}* cx n=3. (**E**) UMAP visualization of intermediate cells in *Pten^{fl/fl}* intact and cx prostates. Left, colored by condition; right, colored by clusters 0–5. (**F**) Relative abundance of intermediate clusters. Top, intact *Pten^{fl/fl}*(n=2); bottom, *Pten^{fl/fl}* cx (n=3) (Data presented as +/-SD). (**G**) Heatmap of composite score for AR, CCP, and Reactome translation gene signatures in intermediate clusters. (**H**) Heatmap of composite score for AR, CCP, and Reactome translation gene signatures in WT intact and castrate basal and luminal cells.

The online version of this article includes the following figure supplement(s) for figure 3:

**Figure supplement 1.** Castration-resistant intermediate cells are phenotypically diverse.

Importantly, proliferation and translation activity scores were inversely correlated with AR signaling, with clusters 1 and 4 exhibiting high proliferation and translation scores and low AR scores (*Figure 3G*). We designated the clusters as AR-high, -medium, or -low and noted that intact *Pten^{fl/fl}* mouse prostates contain mostly AR-high intermediate cells (*Figure 3—figure supplement 1E–F*). Upon castration the number of AR-medium and -low cells increase substantially (*Figure 3—figure supplement 1E–F*). These findings suggest that castration may select for lower AR expressing cell types with high proliferation, but also conserve some 'intact-like' regions with relatively high AR activity and low proliferation. We also conclude that AR activity in intermediate cells may be correlated with cell of origin, as clusters expressing basal markers (clusters 1 and 4) exhibit low AR activity while cluster 3, which expresses luminal markers, has AR activity comparable to differentiated luminal cells (*Figure 3G*, *Figure 3—figure supplement 1B*). Given these correlations, an alternate hypothesis might be that increased basal transdifferentiation, rather than a selection event, is responsible for the increased abundance of AR-low intermediate cells. Lineage-tracing experiments evaluating proliferation and transdifferentiation differences between intact and castrated *Pten^{fl/fl}* mice will be necessary to fully establish a mechanism for increasing AR-low intermediate cell populations.

Our findings raise the question of what aspects of intermediate cell heterogeneity are driven by castration versus loss of *Pten*? To address this question, we performed single-cell RNA sequencing on 9 WT castrated mouse prostates, which we binned into three replicates to overcome small tissue yields due to atrophy in castrated WT prostates. We found that non-basal cells in WT castrate mice aggregate into one clustered region separate from both WT intact luminal cells and *Pten^{fl/fl}* intermediate cells, but interestingly were closest to WT luminal progenitor cells (*Figure 3—figure supplement 1G*). We evaluated the changes in AR activity, translation, and proliferation in basal and luminal cells between WT intact and castrated cells. We found that similar to *Pten^{fl/fl}* intermediate cells, lower AR activity was associated with an increase in a translation signature. However, unlike the *Pten^{fl/fl}* intermediate cells we observed a decrease in the proliferation signature (*Figure 3H*). These findings demonstrate that castration plays an important role in translational heterogeneity in intermediate cells. However, the change in proliferation is likely related to loss of *Pten*.

## Castrate intermediate cell signature correlates with advanced prostate cancer and worse patient outcomes

Given the widespread use of the *Pten^{fl/fl}* model (*Ding et al., 2011*; *Svensson et al., 2011*; *Hsieh et al., 2012*; *Garcia et al., 2014*; *Hsieh et al., 2015*; *Ku et al., 2017*; *Allott et al., 2018*; *Antoch et al., 2020*; *Morel et al., 2021*; *Quaglia et al., 2021*) and our new understanding of cellular dynamics in the context of disease progression, we sought to determine which cell type and context most closely correlated with human prostate cancers that went on to resist androgen deprivation therapy (ADT). To this end, we used a gene signature of ADT resistance derived from human prostate tumors prior to treatment with ADT plus the AR inhibitor enzalutamide, in a neoadjuvant setting (*Figure 4A*) (NCT02430480) (*Karzai et al., 2021*; *Wilkinson et al., 2021*; *Ku et al., 2021*). We generated DEG lists for each epithelial cell type (basal, intermediate, luminal) comparing WT and *Pten^{fl/fl}* or *Pten^{fl/fl}* intact and castrated prostates and performed enrichment analysis using the ADT resistance signature. Out of all the cell types, castrated intermediate cells compared to intact intermediate cells exhibited the most enrichment for the resistance signature (*Figure 4B*). The top five genes from the resistance signature that were upregulated in castrated intermediate cells were *ATP1B1*, *BST2*, *CP*, *IGFBP3*, and *PTTG1*. Importantly, these genes were downregulated in cluster 3 of our intact intermediate cells compared to castrate clusters, demonstrating specificity for aggressive disease (*Figure 4—figure supplement 1A*). Furthermore, all five genes were upregulated in human tumors

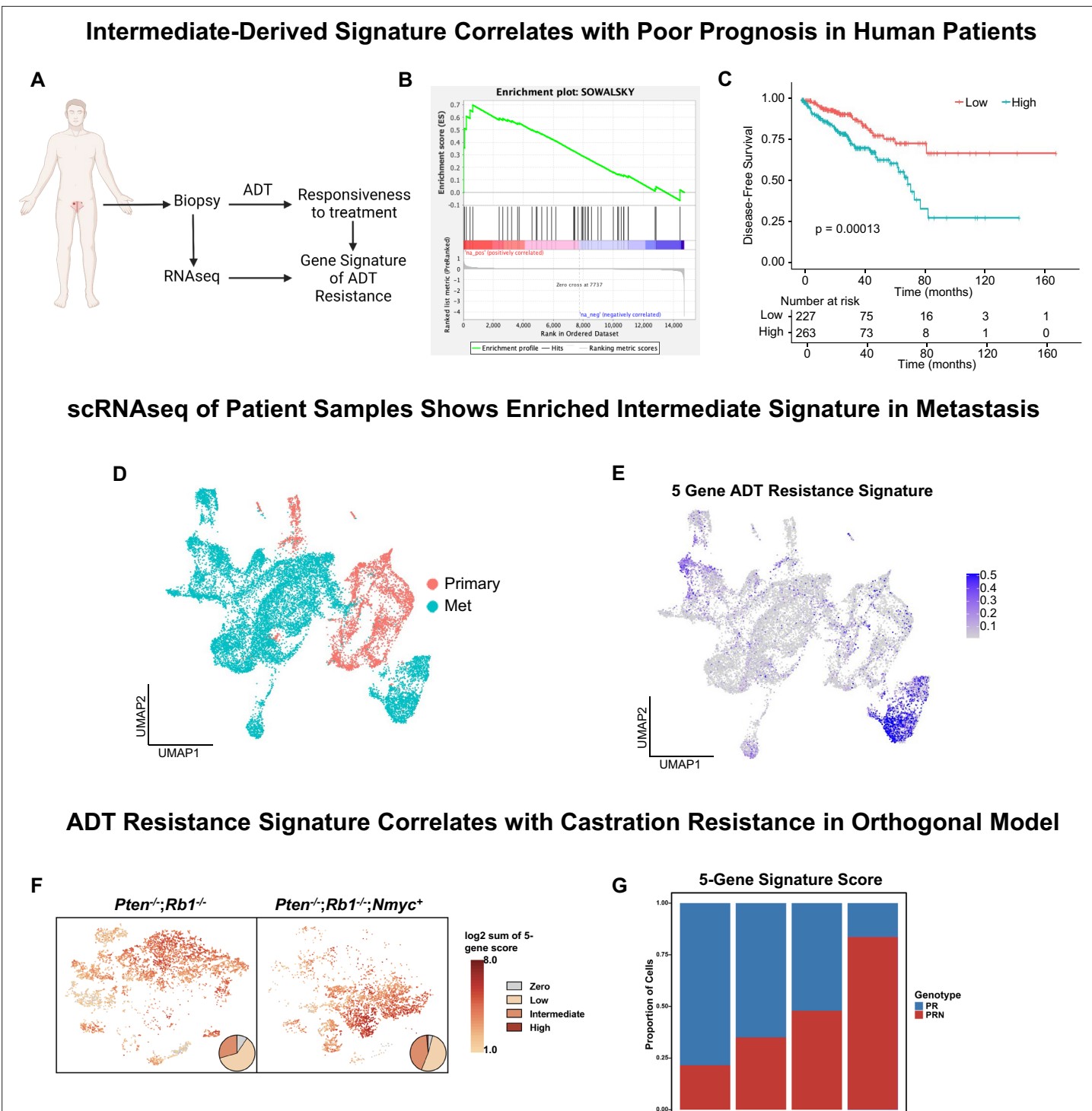

**Figure 4.** Intermediate cells are enriched for a signature of treatment resistance that correlates with advanced human disease. (**A**) Diagram of clinical trial used to establish gene signature of androgen deprivation treatment resistance (NCT02430480). (**B**) Enrichment plot of ADT resistance gene signature relative to intermediate cell DEGs between *Pten*$^{fl/fl}$ and *Pten*$^{fl/fl}$ cx (adjusted p-value = 0.00381). (**C**) Kaplan-Meier curve of disease-free survival for prostate cancer patients in TCGA database with or without high RNA expression of top correlated genes from **B**. Red line, patients with normal expression of all genes; blue line, patients with expression of at least 1 gene with TPM (transcripts per million) in the 80th percentile or above. (**D**) UMAP of tumor cells from human patient samples. Red, primary cancer; blue, metastatic cancer. (**E**) UMAP visualization of per-cell computed score for 5-gene signature from (**B-C**) in human cancer samples. (**F**) UMAP visualization of per-cell computed score for 5-gene signature from (**B-C**) in PR and PRN mouse

*Figure 4 continued on next page*

*Figure 4 continued*

models. Pie charts indicate proportion of cells with zero, low, intermediate, or high signature scores. (**G**) Stacked bar chart showing proportion of cells from PR or PRN mice in each scoring category for the 5-gene signature.

The online version of this article includes the following figure supplement(s) for figure 4:

**Figure supplement 1.** 5 Genes expressed in intermediate cells correlate with poor disease outcomes in human patients.

that exhibited pathologic resistance to ADT (*Figure 4—figure supplement 1B*). Lastly, we sought to investigate whether these genes were associated with worse outcomes for prostate cancer patients. We examined disease-free survival (DFS) of patients stratified by high or low expression of the five top genes across two major prostate cancer cohorts (*Network, 2015*; *Taylor et al., 2010*). We found that patients whose tumor samples expressed high levels of any of these five enriched genes experienced significantly shorter disease-free survival (*Figure 4C*, *Figure 4—figure supplement 1C*). We also analyzed patients with *PTEN* loss or PI3K/AKT pathway dysregulation from the TCGA dataset and found the same trend (*Figure 4—figure supplement 1D*). These results suggest that the castrate intermediate cell state in the *Pten^{fl/fl}* may correlate with poor patient outcomes.

Given our finding that a 5-gene signature derived from castrate intermediate cells correlates to human clinical phenotypes, we investigated if this signature is enriched in metastatic disease. We integrated single-cell RNA sequencing data from primary prostate cancers (n=11) (*Song et al., 2022*) and metastatic prostate cancers (n=6) (*Dong et al., 2020*) and found that primary and metastatic cells clustered separately (*Figure 4D*). We generated a composite score of our 5-gene signature on a per-cell basis and visualized it on the UMAP. We observed an enrichment of the signature only in metastatic patients (*Figure 4E*). Importantly, the cluster containing this signature was composed of cells from five out of the six metastatic samples. These findings show that our 5-gene signature is enriched in human metastatic prostate cancer.

Finally, a possible confounding factor in this model is that physical castration of the *Pten^{fl/fl}* model alone could be driving the expression of the 5-gene signature. To investigate this, we examined single-cell sequencing data from an orthogonal non-castrated mouse model of cancer progression (*Brady et al., 2021*). This study used prostate specific autochthonous models including a double knockout *Pten^{-/-}/Rb1^{-/-}* mouse (PR) and a triple mutant *Pten^{-/-}/Rb1^{-/-}/Nmyc^+* mouse (PRN). PRN mice generate prostate tumors with significantly stronger castration resistance compared to PR mice (*Brady et al., 2021*). Single-cell RNA sequencing data in this study was produced without castration in both PR and PRN mice, resulting in a model of cancer progression outside of physical castration. We overlaid our 5-gene resistance signature on UMAPs of both mouse models and found strong enrichment of the signature in PRN mice compared to PR mice (*Figure 4F–G*). These results reveal that the 5-gene signature is likely not driven by castration alone. Together, our findings demonstrate that a gene signature generated through a specific cell-type within murine prostate cancer closely correlates with human CRPC and worse outcomes in patients.

## Androgen deprivation decreases immune cell abundance but activates *Tnf* signaling

Androgen deprivation induces a host of physiological changes in the prostate, including modulations of immune signaling (*Sha et al., 2015*; *Lopez-Bujanda et al., 2021*). Having observed significant epithelial changes in castrated *Pten^{fl/fl}* mice, we next investigated the consequences of castration on the immune environment. We observed a significant decline in the abundance of all 3 macrophage subtypes, as well as CD8^+ T cells, relative to intact *Pten^{fl/fl}* mice (*Figure 5A–B*). We performed ligand-receptor interaction analysis to understand how androgen deprivation disrupts cell-cell signaling patterns (*Supplementary file 4*). We found that epithelial signaling to macrophage cells was still intact, with relatively little change in ligand or receptor expression (*Figure 5—figure supplement 1A–B*). However, the *Ccl2/7/11-Ccr2* signaling axis from fibroblasts to M2 macrophages and TAMs was entirely ablated in castrated *Pten^{fl/fl}* mice (*Figure 5C*). *Ccr2* expression was significantly decreased in macrophages, and *Ccl2/7/11* were all dramatically decreased in fibroblasts (*Figure 5D*). This suggests that fibroblast-mediated macrophage recruitment is interrupted by androgen deprivation. Indeed, androgen signaling is known to promote pro-tumorigenic macrophage function as well as macrophage recruitment via *Ccr2* expression, lending credence to this hypothesis (*Lai et al., 2009*;

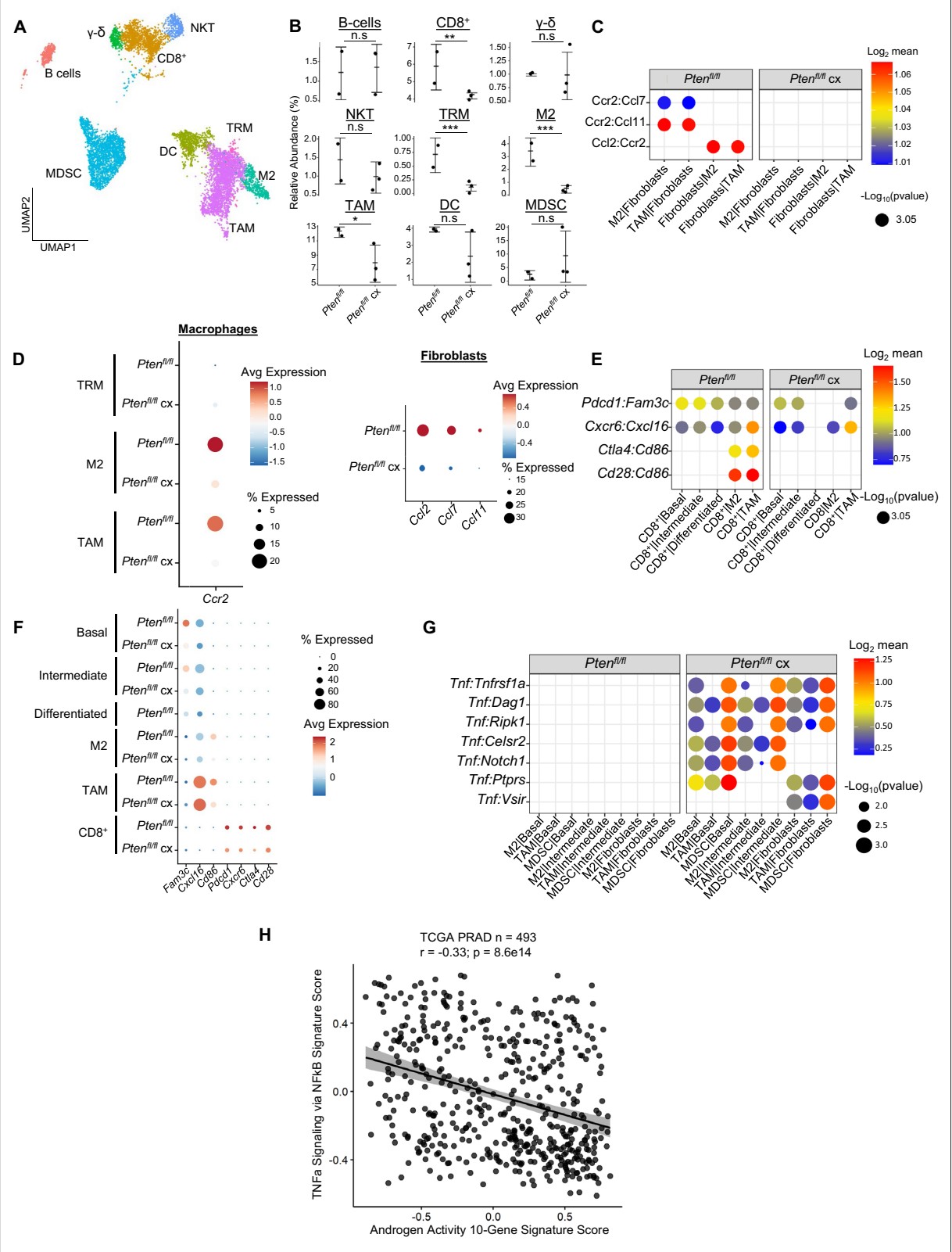

**Figure 5.** Castration remodels immune environment via fibroblast signaling and increases TNF pathway activity. (**A**) Combined UMAP visualization of immune cells in *Pten*^fl/fl^ and *Pten*^fl/fl^ cx ventral prostates. (**B**) Relative abundance of immune cells in *Pten*^fl/fl^ intact (n=2) and cx (n=3) mice. Y-axis shows the % composition of each sample by cell type (Data presented as +/-SD, *p<0.05, **p<0.01, ***p<0.001, n.s=not significant, negative binomial test). (**C**) Dot plot of signaling interactions between macrophages and fibroblasts. (**D**) Dot plot of *Ccr2* expression in M2 macrophages and TAMs (left). Dot plot

*Figure 5 continued on next page*

*Figure 5 continued*

of *Ccr2* ligand expression in fibroblasts in *Pten^fl/fl* intact and cx mice (right). (**E**) Dot plot of signaling interactions between CD8⁺ T cells and epithelial and macrophage cells in *Pten^fl/fl* intact and cx mice. (**F**) Dot plot of epithelial and macrophage ligands and CD8⁺ T cell receptor gene expression in *Pten^fl/fl* intact and cx mice. (**G**) Dot plot of *Tnf* signaling interactions between myeloid and epithelial/fibroblast cells in *Pten^fl/fl* intact and cx prostates. (**H**) Scatter plot of TCGA PRAD study patient signature composite scores. Y-axis, TNF signaling signature score; X-axis, AR signaling signature score (Pearson's correlation).

The online version of this article includes the following figure supplement(s) for figure 5:

**Figure supplement 1.** Epithelial-mediated macrophage recruitment is not interrupted by castration.

*Cioni et al., 2020*; *Becerra-Diaz et al., 2020*). TRMs also decreased in abundance, but they were not targeted by fibroblast signaling and epithelial signaling was uninterrupted. Given the expression of AR-dependent genes in this macrophage subtype (*Figure 2—figure supplement 1D*), we speculated that loss of androgen signaling could be deleterious to this population. Indeed, in intact *Pten^fl/fl* mice TRMs had high AR activity relative to other macrophage subtypes, and this activity was decreased by castration (*Figure 5—figure supplement 1C*). This suggests that castration interrupts fibroblast-mediated pro-tumorigenic M2 and TAM recruitment and depletes the androgen-dependent tissue-resident macrophage reservoir, leading to decreased macrophage abundance in the prostate.

Macrophages likely contribute to CD8⁺ T cell recruitment in intact *Pten^fl/fl* mice (*Figure 2G–H*); we speculated that macrophage-mediated signaling might be interrupted in the context of castration and cause a decrease in CD8⁺ T cell abundance. Indeed, while epithelial-CD8⁺ interactions were mostly intact, *Cd86-Cd28/Ctla4* signaling from M2 macrophages and TAMs was disrupted, and *Cd86* expression was greatly reduced in both M2s and TAMs (*Figure 5E*). In addition, receptor expression was decreased in CD8⁺ T-cells, including suppressive markers *Ctla4* and *Pdcd1* (*Figure 5F*). These findings suggest that depletion of the macrophage population causes a decrease in both CD8⁺ T cell recruitment and suppression, possibly leading to more cytotoxic but less abundant CD8⁺ T cells in castrated *Pten^fl/fl* mice.

Finally, *Tnf* signaling has previously been implicated as a pro-tumorigenic factor in AR low prostate cancer (*Mizokami et al., 2000*; *Sha et al., 2015*). Accordingly, we examined *Tnf* interactions in our ligand-receptor analysis and found a striking enrichment of *Tnf* pathway activity in castrated mice. Specifically, pro-tumorigenic myeloid cells (M2 macrophages, TAMs, and MDSCs) expressing *Tnf* interact with multiple receptors in epithelial cells and fibroblasts (*Figure 5G*). To investigate whether this association held true in human prostate cancer, we correlated a 200 gene signature of *Tnf* activity (*Griss et al., 2020*) with AR signaling activity in the prostate cancer TCGA dataset (*Network, 2015*). We found a significant inverse correlation between TNF and AR activity in human patients, validating our finding that TNF signaling is induced in prostate cancer upon castration (*Figure 5H*). This correlation also held true when only considering patients with PTEN loss (*Figure 5—figure supplement 1D*). We conclude that castration in the *Pten^fl/fl* mouse provokes several large-scale cellular signaling changes that result in decreased macrophage and CD8⁺ T cell populations and increased *Tnf* signaling.

## Translation inhibition in AR-low prostate cancer is lethal to basal and intermediate cells and disrupts pro-tumorigenic signaling

Deregulated mRNA translation rates have previously been implicated in aberrant gene expression and aggressive AR-low prostate cancer in the *Pten^fl/fl* mouse (*Liu et al., 2019*; *Lim et al., 2021*). However, understanding how the per cell requirement for aberrant translation enables tumor heterogeneity has been technically challenging. Therefore, it remains to be determined which prostate cancer epithelial cell types require increased translation for androgen independent growth. Given the strong correlation between proliferation and translation observed in both basal and intermediate cells (*Figure 3*, *Figure 3—figure supplement 1D*), we hypothesized that inhibiting translation in the *Pten^fl/fl* mouse could be deleterious to both cell types. To investigate this possibility, we used the *PB-Cre4;Pten^fl/fl;ROSA26-rtTA-IRES-eGFP;TetO-*4ebp1^M mouse model (herein referred to as *Pten^fl/fl;*4ebp1^M). In this model, Cre-mediated recombination leads to *Pten* loss and expression of the rtTA protein in both basal and intermediate cells (*Figure 1—figure supplement 1B*). When mice are treated with doxycycline, a mutant *Eif4ebp1* transgene (4ebp1^M) is expressed (*Hsieh et al., 2015*). eIF4EBP1 is a negative regulator of translation initiation and functions via inhibition of eIF4F complex assembly (*Schuster and Hsieh, 2019*). This mutant allele cannot be inactivated via mTOR-mediated phosphorylation and

its expression robustly inhibits eIF4F complex formation and translation initiation in prostate epithelia (*Figure 6A*). Using this model, we sought to determine the epithelial cell type specific dependencies of castration resistant prostate cancer. Here, doxycycline was administered starting at 4 months of age, simultaneously with castration, and prostates were collected at 6 months of age.

Initially, no major differences were observed in the UMAP comparing all epithelial cells in castrated *Pten*<sup>fl/fl</sup> mice with or without 4ebp1<sup>M</sup> (*Figure 6B*). However, we noted a striking (~10-fold) depletion of the *rtTA-eGFP* transgene which is required for 4ebp1<sup>M</sup> induction in *Pten*<sup>fl/fl</sup>;4ebp1<sup>M</sup> basal and intermediate epithelial cells compared to the *Pten*<sup>fl/fl</sup> model (*Figure 6C*, *Table 1*). We hypothesized that translation inhibition in AR-low prostate cancer might be lethal or confer a competitive disadvantage to AR low epithelial cells, and that the bulk of the remaining epithelia did not express the 4ebp1<sup>M</sup> and were simply castrate cells that did not express the transgene. To test this hypothesis, we performed DEG analysis only on transgene-positive cells, comparing castrated cells with or without 4ebp1<sup>M</sup>. We found many more differentially expressed genes between these groups in basal cells (465 differentially expressed genes) than in the non-filtered analysis (56 DEGs) (*Supplementary file 5A-B*). We did not observe a significant change in the number of DEGs between castrated *Pten*<sup>fl/fl</sup> and *Pten*<sup>fl/fl</sup>;4ebp1<sup>M</sup> intermediate cells when filtering for *rtTA-eGFP+* cells. This may be due to the high phenotypic diversity in this compartment and the very low proportion of transgene-positive intermediate cells after 4ebp1<sup>M</sup> induction (<1%, *Table 1*) causing a lack of robustness in the DEG analysis.

Interestingly, upon performing pathway analysis on basal cell DEGs we found enrichment of pathways relating to translation, cell cycle arrest, and apoptosis, and observed downregulation of mitochondrial function and mTORC1 signaling pathways (*Figure 6D*). These findings suggest that remaining transgene-positive basal cells in the *Pten*<sup>fl/fl</sup>;4ebp1<sup>M</sup> mice may be undergoing cell cycle arrest, interruption of growth processes such as inhibition of the mTOR pathway, and apoptosis. We also noted that in the basal compartment, the proportion of hyperproliferative cells had decreased drastically (*Table 2*). This suggests that highly proliferative basal cells are more dependent upon high translation than less proliferative basal cells.

Given the large-scale changes in epithelial populations caused by 4ebp1<sup>M</sup> induction in castrated *Pten*<sup>fl/fl</sup> mice, we next asked how cell-cell signaling between the remaining epithelial cells and other compartments were affected by the loss of basal and intermediate cell populations through translation inhibition (*Supplementary file 5C*). We found that *Tnf* signaling was greatly decreased between myeloid and epithelial cells (*Figure 6E–F*). In addition, epithelial-fibroblast and inter-epithelial *Egfr* signaling also decreased significantly (*Figure 6G–H*). *Egfr* activity is associated with worse cancer prognosis, and inhibition of *Egfr* signaling has been proposed as a therapeutic approach in advanced prostate cancer (*Kim et al., 2006*; *Guérin et al., 2010*; *Xiong et al., 2020*). Overall, these findings show that both basal and intermediate cancer cell types require increased translation initiation to maintain castration resistance and that aberrant mRNA translation is required for tumor heterogeneity. Furthermore, translation inhibition of distinct epithelial populations can impact the local tumor microenvironment.

## Discussion

Here, we demonstrated that *Pten* deletion in mouse prostate epithelial cells results in the generation of an intermediate luminal subtype, which is phenotypically similar to but likely distinct from the *Psca+/Krt4+/Tacstd2+* luminal progenitor populations in the WT prostate. In this model, the *Pten*<sup>fl/fl</sup> intermediate cells likely derive from three cellular pools: basal cells, luminal progenitor cells, and differentiated luminal cells. Our findings support the idea that basal cells can transdifferentiate into intermediate cells upon loss of *Pten*. This corresponds well with lineage tracing studies (*Choi et al., 2012*; *Wang et al., 2013*; *Lu et al., 2013*) that showed that basal cells can be cells of origin for prostate cancer and transition to luminal phenotypes during transformation in mice. We also identify common luminal markers expressed in some intermediate cells, suggesting luminal cells may also phenotype switch to an intermediate state upon *Pten* loss. Lastly, the close proximity of luminal progenitor cells to intermediate cells and the fact they can express AR and form neoplasia in the context of *Pten* loss (*Guo et al., 2020*) reveal another potential source of intermediate cells. Our study is limited by the fact that the Cre driver is active in all epithelial subtypes. To observe transdifferentiation or phenotype switching, cell type specific lineage tracing will be required. New techniques such as DNA Typewriter

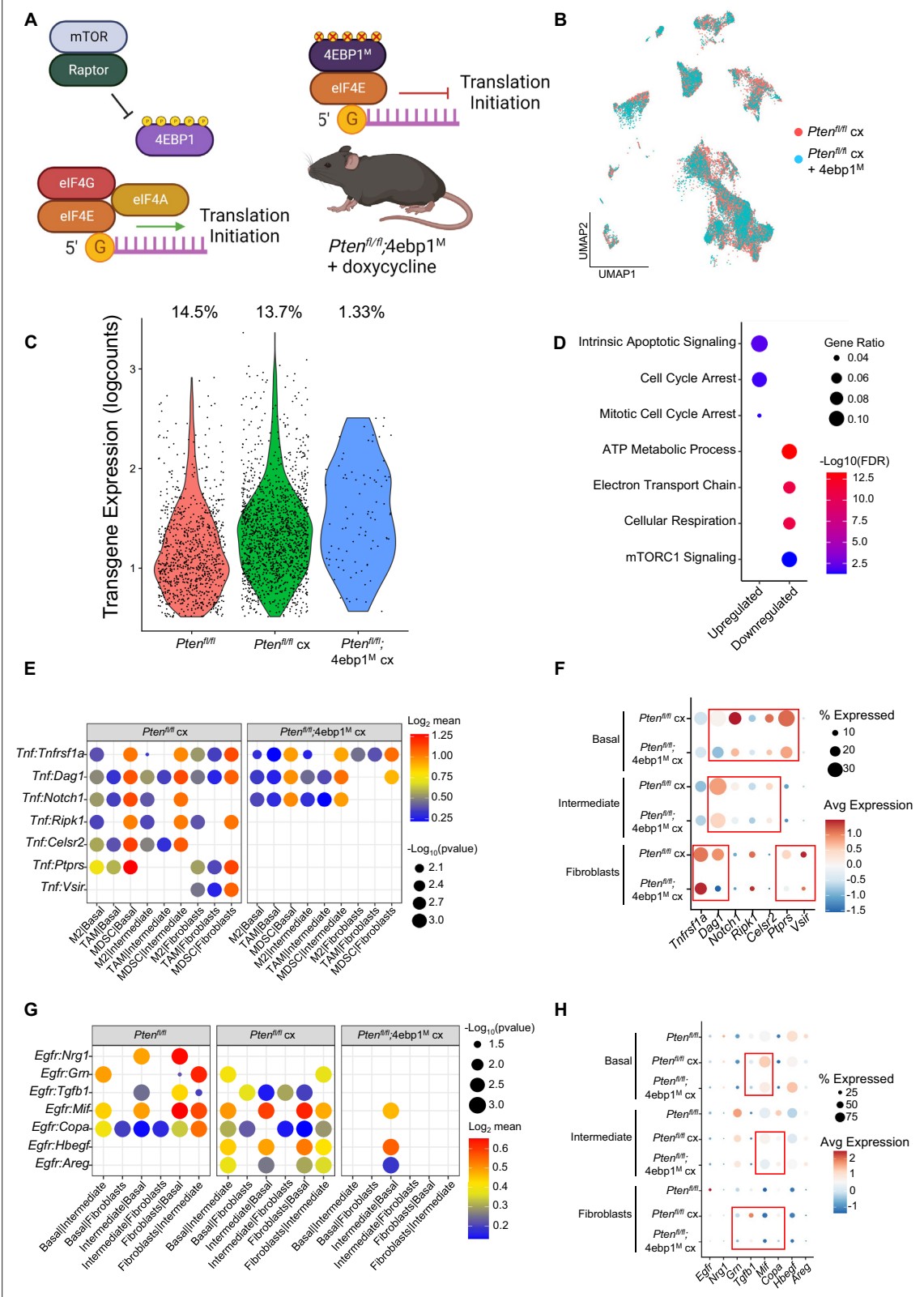

**Figure 6.** 4EBP1[M] expression is lethal in epithelial cells and decreases EGFR and TNF ligands in epithelial cells and fibroblasts. (**A**) Simplified schematic of the eIF4F translation initiation complex and how the 4EBP1[M] protein functions in the *Pten^{fl/fl};4ebp1^M* mouse model when treated with doxycycline. (**B**) UMAP visualization of epithelial cells in *Pten^{fl/fl}* cx and *Pten^{fl/fl};4ebp1^M* cx prostates, colored by genotype. (**C**) Violin plot of *rtTA-eGFP* transgene expression in epithelial cells in each *Pten^{fl/fl}* condition. Plot shows only cells expressing the transgene; each dot represents a cell. Percentages represent

*Figure 6 continued on next page*

*Figure 6 continued*

the proportion of transgene-positive cells in each condition. (**D**) Dot plot of top GSEA results from DEG analysis of transgene-positive basal cells in *Pten*<sup>fl/fl</sup>;*4ebp1*<sup>M</sup> cx mice compared to *Pten*<sup>fl/fl</sup> cx ventral prostates. All pathways are enriched with FDR <0.05. (**E**) Dot plot of *Tnf* signaling interactions between myeloid and epithelial/fibroblast cells in *Pten*<sup>fl/fl</sup> cx and *Pten*<sup>fl/fl</sup>;*4ebp1*<sup>M</sup> cx mice. (**F**) Dot plot of *Tnf* and *Tnf* ligand expression in myeloid cells, epithelial cells, and fibroblasts in *Pten*<sup>fl/fl</sup> cx and *Pten*<sup>fl/fl</sup>;*4ebp1*<sup>M</sup> cx prostates. Red boxes highlight ligands with decreased expression in *Pten*<sup>fl/fl</sup>;*4ebp1*<sup>M</sup> cx mice. (**G**) Plot of *Egfr* signaling interactions between epithelial cells and fibroblasts in *Pten*<sup>fl/fl</sup> intact, *Pten*<sup>fl/fl</sup> cx, and *Pten*<sup>fl/fl</sup>;*4ebp1*<sup>M</sup> cx prostates. (**H**) Dot plot of *Egfr* and *Egfr* ligand expression in epithelial cells and fibroblasts in *Pten*<sup>fl/fl</sup> intact, *Pten*<sup>fl/fl</sup> cx, and *Pten*<sup>fl/fl</sup>;*4ebp1*<sup>M</sup> cx ventral prostates. Red boxes highlight ligands with decreased expression in *Pten*<sup>fl/fl</sup>;*4ebp1*<sup>M</sup> cx mice.

The online version of this article includes the following figure supplement(s) for figure 6:

**Figure supplement 1.** Interactive Portal, Enabling Gene- and Cell- Specific Comparisons Across the Spectrum of Prostate Cancer Initiation and Progression in vivo.

---

may offer flexible multiplexed platforms to track multiple cell lineages in the same system simultaneously (**Choi et al., 2022**).

We observed that intermediate cells undergo significant cell state changes upon castration which increases their heterogeneity. This heterogeneity is characterized by a spectrum of AR signaling inversely correlated with both proliferation and translation activity. Within this spectrum, clusters with high AR activity and low proliferation and translation express some luminal markers, while low AR, high proliferation, high translation clusters express basal markers. This finding supports the hypothesis that both basal and luminal cells can transdifferentiate to intermediate cells in cancer and suggests that basal-originating intermediate cells may represent more aggressive cancer phenotypes. These specific intermediate clusters may be responsible for the increased tumor growth previously observed at the whole-prostate level in castrated *Pten*<sup>fl/fl</sup> mice (**Liu et al., 2019**). This lends importance to the question of whether increased AR-low intermediate populations in *Pten*<sup>fl/fl</sup> castrate mice are due to the expansion of an existing cell state, or renewed lineage plasticity events from basal cells. Comparative lineage tracing experiments measuring basal transdifferentiation and proliferation dynamics in intact and castrated *Pten*<sup>fl/fl</sup> mice will be necessary to fully address this question. Overall, we speculate that the intermediate cell compartment is highly heterogeneous in *Pten*<sup>fl/fl</sup> mice partly due to multiple cellular origins converging on a common phenotype.

We further characterize the effects of intermediate cell expansion by showing that a 5-gene signature enriched in this cell type correlates with poor patient outcomes, advanced human CRPC, and castration resistance in an orthogonal mouse model. Specifically, we observe that upregulation of these 5 genes significantly correlates with shorter disease-free survival in two datasets of bulk RNA-seq from patients. We also find that the 5-gene signature is specifically enriched in a subset of metastatic, but not primary, human prostate cancer cells (**Song et al., 2022**; **Dong et al., 2020**). Finally, we demonstrate the enrichment of the signature in an orthogonal murine model of prostate cancer progression and castration resistance (**Brady et al., 2021**). Our findings suggest that intermediate cells are enriched for a gene signature of treatment resistance in human tumors and pose the intriguing question of whether detection of this population in patients can provide a useful prognostic tool. Further in vivo studies will be necessary to validate the role of intermediate cell states in human disease and their potential implications in clinical medicine.

The prostate tumor environment is typically described as immunosuppressive and does not readily respond to immunotherapies (**Kantoff et al., 2010**; **Cham et al., 2020**; **Dong et al., 2020**). Correspondingly, we show that the immune environment of the *Pten*<sup>fl/fl</sup> mouse prostate is highly enriched

**Table 1.** Transgene abundance in PTEN mouse epithelia.

| Cell Type | | *Pten*<sup>fl/fl</sup> | *Pten*<sup>fl/fl</sup> cx | *Pten*<sup>fl/fl</sup>;*4ebp1*<sup>M</sup> |
|---|---|---|---|---|
| Basal | All | 1795 | 2718 | 1565 |
| | *rtTA-eGFP+* | 255 (14.2%) | 422 (15.5%) | 41 (2.6%) |
| Intermediate | All | 2391 | 8230 | 4760 |
| | *rtTA-eGFP+* | 438 (18.3%) | 1073 (13.0%) | 43 (0.90%) |

**Table 2.** Basal proliferative subset proportions in *rtTA-eGFP+* cells.

| Basal Subset | Hypo-proliferative | Hyper-proliferative |
|---|---|---|
| *Pten*$^{fl/fl}$ | 188 (73.7%) | 60 (23.5%) |
| *Pten*$^{fl/fl}$ cx | 313 (74.1%) | 98 (23.2%) |
| *Pten*$^{fl/fl}$;4ebp1$^M$ | 38 (97.4%) | 1 (2.6%) |

for pro-tumorigenic immune cells, specifically immunosuppressive myeloid cells and exhausted CD8+ T cells. We carefully delineated cell-cell signaling patterns and found that epithelial cells, fibroblasts, and macrophages contribute to immune recruitment. Since M2 and tumor-associated macrophages most significantly contribute to recruitment of other immune cells, macrophages are likely activated and recruited first during tumorigenesis, and subsequently emit chemokines and other ligands that help attract and exhaust CD8+ T cells as well as pro-tumorigenic MDSCs. Interestingly, specific macrophage subtypes seem differentially responsible for distinct recruitment patterns: M2 macrophages express very high levels of MDSC-associated chemokines, while TAMs recruit CD8+ T cells via high expression of *Cxcl16* and *Cd86*. Given these results, interrupting the recruitment of tumor-associated macrophages may be a valid strategy for depleting pro-tumorigenic immune populations and overcoming immunotherapy resistance in prostate cancer. In vitro studies utilizing co-cultures of epithelial and immune cells will be required to validate these in silico results and confirm whether the epithelia-macrophage signaling axis is an appropriate therapeutic target.

Androgen deprivation in the *Pten*$^{fl/fl}$ mouse leads to a decrease in macrophage and CD8+ T cell abundance, likely due to ablation of fibroblast-mediated chemokine signaling. Androgen signaling is active in tissue-resident macrophages and can induce pro-tumorigenic behaviors including increased migration and proliferation in prostate cancer cells (*Cioni et al., 2020*). In addition, AR can promote *Ccr2* expression and facilitate macrophage activity and recruitment (*Lai et al., 2009*). This corresponds well to our finding that fibroblast-mediated signaling towards *Ccr2* contributes to macrophage recruitment and is interrupted upon castration, and suggests that androgen deprivation decreases macrophage abundance in the prostate tumor environment. Conversely, castration increased *Tnf* signaling from myeloid cells to epithelial cells and fibroblasts. TNF signaling has been described as pro-tumorigenic in AR-low prostate cancer (*Mizokami et al., 2000*; *Sha et al., 2015*), and we confirmed that TNF activity was inversely associated with AR activity in human patients. This paradigm supports the role of macrophages and neutrophils in maintaining a favorable tumor environment even in the context of androgen loss and suggests a mechanistic relationship between castration and pro-tumorigenic immune signaling.

Finally, we tested the hypothesis that high translation rates were important to maintain AR-low prostate cancer heterogeneity using the *Pten*$^{fl/fl}$;4ebp1$^M$ mouse model. We discovered that 4ebp1$^M$ induction severely depleted both basal and intermediate cells. Interestingly, hyper-proliferative basal cells were preferentially depleted, leading to a decrease in overall basal cell proliferation. In addition, *Egfr* and *Tnf* signaling were decreased in the tumor microenvironment. We conclude that high translation rates are essential to maintain tumor heterogeneity in AR-low prostate cancer and may play a role in pro-tumorigenic cell-cell communication pathways. Based on these findings, we speculate that translation inhibitors may represent a therapeutic modality to decrease tumor heterogeneity, and that additional studies may uncover further druggable targets. Overall, our work highlights multiple epithelial and immune cell types crucial to prostate cancer initiation and progression and elucidates interactions between specific cell populations that may facilitate castration resistance. Lastly, this work aims to provide a broad, searchable resource to the cancer research community. To this end, we have developed a publicly accessible and interactive website (available at https://atlas.fredhutch.org/hsieh-prostate/) that allows for cell- and gene-specific queries through all 50,780 cells analyzed in this study (*Figure 6—figure supplement 1*).

# Materials and methods

## Mice

*PB-Cre4* mice which express a *Cre* recombinase transgene under the control of the rat *Pbsn* (probasin) promoter in the prostate epithelium were obtained from the Mouse Models of Human Cancer Consortium (now Jax #026662). *Pten^fl/fl* (Jax #006440) and *Rosa-LSL-rtTA* (Jax #005670) mice were obtained from the Jackson Laboratory. *TetO*-4ebp1^M mice which express an inducible mutant *Eif4ebp1* transgene were generated as previously described (*Hsieh et al., 2010*). All mice were maintained in the C57BL/6 background under specific pathogen–free conditions, and experiments conformed to the guidelines as approved by the Institutional Animal Care and Use Committee of Fred Hutchinson Cancer Research Center.

## Surgical castration and activation of the 4EBP1^M transgene

Surgical castrations were performed in 4- to 5-month-old mice under isoflurane anesthesia. Postoperatively, mice were monitored daily for 5 days. To activate the 4EBP1^M transgene, doxycycline (Sigma-Aldrich) was administered in the drinking water at 2 g/liter immediately after castration, and euthanasia was performed 8 weeks after castration.

## Tissue dissociation for single-cell RNA sequencing

Ventral prostate lobes from C57BL/6 J WT, *Pten^fl/fl*, *Pten^fl/fl*;4ebp1^M mice were dissected, washed with chilled 1 X PBS, and then minced with a scalpel into small pieces (~1 mm) in a petri dish. Paired lobes from a single mouse were collected and dissociated into one sample for scRNA-seq. The tissue was digested with DMEM/F12/Collagenase/Hyaluronidase/FBS (StemCell technologies, Vancouver, Canada) for 1 hr at 37 °C on a slowly shaking/rotating platform. The tissue was further digested in 0.25% Trypsin-EDTA (Invitrogen, Carlsbad, CA) on ice for 30 minutes, and followed by suspension in Dispase (Invitrogen, Carlsbad, CA, 5 mg/mL) and DNase I (Roche Applied Science, Indianapolis, IN, 1 mg/mL). Any cell clumps were dissociated by gently pipetting up and down. The dissociated cells were then passed through 70 µm cell strainers (BD Biosciences, San Jose, CA) to achieve single cell suspension. The suspension was resuspended with 3 ml PBS (Life Technologies) with 2% fetal calf serum (FCS) (Gemini Bioproducts, West Sacramento, CA) and immediately placed on ice. Viable cells were counted by Vi-Cell XR Cell Viability Analyzer (Beckman Coulter, Brea, CA) and then diluted accordingly to reach the targeted cell concentration.

## Single-cell RNA sequencing library preparation

3′ single-cell RNA libraries were generated according to the protocol outlined in Single Cell 3′ Reagent Kits v2 User Guide 10 X Genomics. Briefly, cells and reverse transcription reagents were partitioned into oil-based Gel Beads in Emulsion (GEMs), with each GEM containing a unique 10 x barcode. Cells were then lysed and underwent reverse transcription resulting in barcoded cDNA. The cDNA was then collected and amplified prior to undergoing library construction in which P5, P7, and a unique sample index were added.

At least two mouse prostates were prepared and sequenced for each condition in order to obtain >10,000 cells/condition, resulting in n=3 for WT mice, n=2 for *Pten^fl/fl* mice, n=3 for *Pten^fl/fl* mice, and n=2 for *Pten^fl/fl*;4ebp1^M mice. Individual mouse prostates were considered biological replicates. The 10 libraries generated in this manner were pooled and sequencing was performed on an Illumina NovaSeq 6000 using the v1.5 S1-100 flow cell and reagent kit. Sequencing configuration was paired-end 26x8 × 96 and Illumina RTA version v3.3.5 was used. This generated a median of 58.7 million reads/sample with a median 54.60% saturation, median 90.70% Q30 fraction, 13,746 average reads/cell, and 6.2 reads/UMI.

## Alignment and filtering of reads

Two transgene transcripts (*Cre*, *rtTA-eGFP*) were added to the mm10 transcriptome to detect transgenes expressed in the *Pten^fl/fl* and *Pten^fl/fl*;4ebp1^M mice. Kallisto v0.45.1 (*Bray et al., 2016*) was used to demultiplex all samples into FASTQ files and align reads to the modified mm10 transcriptome. DropletUtils package (*Lun et al., 2019*; *Griffiths et al., 2018*) was used to filter out empty or duplexed cell droplets. Cells with fewer than 200 or greater than 5000 detected genes, fewer than

500 or greater than 25,000 detected UMIs and cells with >15% mitochondrial reads were filtered from subsequent analysis.

## PCA, UMAP, and clustering

R package Seurat v4.0.4 (https://satijalab.org/seurat/) was used to construct a Principal Component Analysis (PCA) for the entire dataset using the 2000 most variable genes as features. The Uniform Manifold Approximation and Projection (UMAP) dimension reduction technique was used for visualization and the R function 'FindClusters()' with resolution = 0.2 was used to generate 43 clusters.

## Cell type identification

The SingleR package v1.6.1 was used to assign initial cell type identities to each cluster. These IDs were verified and refined using expression patterns of published biomarkers. For epithelial cells, cell subtypes (basal, intermediate, differentiated) were assigned using published gene signatures from other single-cell RNA sequencing projects. For immune cells, broad cell types (T cells, macrophages) were divided into activation states via known biomarkers (e.g. *Cd8a* for CD8 T cells and *Mrc1* for M2-activated macrophages). Stromal cell types were also determined via biomarkers.

## Relative cell abundance

To compare the abundance of specific cell populations while controlling for sample library size, the percentage composition of each sample was calculated by cell type. Statistical significance was generated via a negative binomial regression test to determine whether a given cell type was over- or under-represented between conditions.

## Gene signature enrichment

The GSVA package v1.40.1 (*Hänzelmann et al., 2013*) was used to generate composite scores for gene signatures such as a 20-gene AR activity signature or the 30-gene CCP proliferation signature. Due to the sparse nature of single-cell transcriptomes, the data was pseudo-bulked by sample and cell type to generate more robust analyses. Statistical analysis was performed via permutation test with 10,000 permutations.

## Differential expression and gene set enrichment analysis

Differential gene expression was computed using Seurat functions with a threshold log2 fold-change >0.25 or<–0.25 and FDR <0.05. Upregulated and downregulated genes were further filtered by setting a log2 fold-change threshold = log2(1.25) = ~0.32. Gene names were converted from mouse to human via the biomaRt package (*Durinck et al., 2009*) and GSEA was performed using the MsigDB database with the C2, C5, C6, C7, Hallmark, KEGG, BioCarta, and Reactome gene sets. Resulting enriched pathways were filtered via a threshold of FDR <0.05.

## Trajectory, RNA velocity, and pseudotime

Monocle3 (*Cao et al., 2019*; *Qiu et al., 2017*; *Trapnell et al., 2014*) and velocyto (*La Manno et al., 2018*) were used to draw trajectory paths and RNA velocity maps, respectively, through the epithelial compartment of the *Pten^{fl/fl}* intact mice. Palantir (*Setty et al., 2019*) was used to delineate gene expression dynamics across pseudotime in basal and intermediate cells in *Pten^{fl/fl}* intact mice.

## Ligand-receptor interactions

Ligand-receptor interactions between cell types were determined via the CellphoneDB package v2.0.0 (*Efremova et al., 2020*). Only interaction with p-value <0.05 were included in the final analysis.

## Cell cycle assignment

Cell cycle phases for single cells were determined using the Seurat cell cycle function, which includes gene lists denoting the G2M and S phases. Gene names were converted from human to mouse using the biomaRt package to match our data, then the CellCycleScoring function was used to assign each cell either S, G2M, or G1 phase. Chi-squared test was used to determine whether the proportions of G1 cells were significantly different between clusters or conditions.

## Human gene signature of ADT resistance and correlation to mouse data

Tumor samples were laser capture microdissected from prostate cancer biopsies prior to undergoing six months of neoadjuvant androgen deprivation therapy plus enzalutamide and ranked based on volume of residual tumor in each patient, as previously described (*Karzai et al., 2021*; *Wilkinson et al., 2021*; *Ku et al., 2021*). Separately, differentially expressed genes (DEGs) derived from the PTEN null intact and castrate basal or intermediate cells were converted from mouse to human gene symbols using getLDS function from the biomaRt package v2.48.3 for R/Bioconductor (*Durinck et al., 2009*). Gene set enrichment analysis (GSEA) was performed on the basal vs. intermediate DEGs set against the top 50 genes associated with treatment resistance, and the top five leading edge genes from GSEA were used to stratify samples. Survival analysis was performed using the *survival* package in R on the TCGA prostate adenocarcinoma (*Network, 2015*) (n=490) and MSKCC (*Taylor et al., 2010*) (n=140) datasets. A cancer sample was considered "altered" if the expression of at least one of the five leading edge genes was greater than the 80th percentile for the entire cohort (TCGA or MSKCC, respectively).

## Integration of single-cell data from human patients

Sequencing data from human primary tumor samples was contributed by Dr. Franklin Huang. Data from human metastatic tumor samples was obtained from the GEO repository of *Dong et al., 2020*. All primary and metastatic samples were processed individually via the standard Seurat workflow. The primary sample object was filtered to only retain confirmed malignant cells. All objects were integrated into a Seurat object using a standard workflow (available here) and visualized via the UMAP dimension reduction method. The AddModuleScore function was used to compute the 5-gene signature score for every cell and was visualized via the FeaturePlot function.

## Orthogonal mouse model

Data from individual $Pten^{-/-}/Rb1^{-/-}$ (PR) and $Pten^{-/-}/Rb1^{-/-}/Nmyc^{+}$ (PRN) mice were aggregated using the cellranger aggr pipeline and visualized using Loupe Cell Browser. Quantification of gene expression signatures was performed using the sum of log 2 -transformed normalized UMI counts across all genes in the signature. Cells were stratified into equal-sized tertiles (low, intermediate, high) based on the maximum value of the signature score. Cells with no detectable UMI counts of any signature genes were assigned to the zero category.

## TCGA analysis of TNF activity

The Cancer Genome Atlas (TCGA) PRAD cohort containing 493 primary prostate tumor samples with RNA-seq expression values was utilized for analysis of signature scores. We used the RSEM values hosted by the cBioPortal (http://www.cbioportal.org, study: prad_tcga_pan_can_atlas_2018.) Single sample enrichment scores were calculated using GSVA (*Hänzelmann et al., 2013*) with default parameters using genome-wide log2 RSEM values as input. The pathways used were from MSigDBv7.4 (HALLMARK_TNFA_SIGNALING_VIA_NFKB) and the 10-gene androgen-regulated (AR) signature from *Bluemn et al., 2017*. In analyses restricted to samples with PTEN biallelic loss, 94 samples were used which had either 2 copy loss or 1 copy loss and a non-synonymous mutation annotated as a putative driver mutation in cBioPortal. Pearson's correlation coefficient was used to study the relationships between signature scores shown in scatterplots using the cor.test function in R.

## Data and code availability

The single-cell RNA sequencing data files are available on the GEO database at GSE171336 and can be accessed using token: ijmfokccrhepvub. The code used to process and analyze the data is available at https://github.com/sonali-bioc/GermanosProstatescRNASeq/, (copy archived at swh:1:rev:5a376d7b-77d034e9bd09ce4787337ee33fda8448; *Arora, 2022*). All other data associated with this study are present in supplementary materials and tables.

## Interactive website

The web-based data Atlas was developed utilizing open-source technologies, including React for the application framework, Material UI for interface components, and Apache EChart for visualizations. All data were extracted from Seurat HDF5 files into web-optimized CSV, Arrow, and Binary files. All site

data and assets are stored in Amazon S3 and served through Amazon CloudFront, a global content delivery network (CDN) service built for high-speed, low-latency performance and security. The site is hosted at https://atlas.fredhutch.org/hsieh-prostate/.

## Acknowledgements

We are grateful to the patients who participated in this study and their families. We thank members of the ACH laboratory for helpful advice and discussions. We thank the Seattle RNA Metabolism group for critical discussion of the work. We thank L Xin for sharing RNA-seq data and critical reading of the manuscript. This work was supported by NIH award R37 CA230617, R01 GM135362, the Pacific Northwest Prostate Cancer SPORE DRP (P50 CA097186), Burroughs Welcome Fund, Career Award for Medicine Scientists (1012314.02), and grants from the Emerson Collective (691630), and the Robert J Kleberg Jr. and Helen C Kleberg Foundation to ACH. AAG received funding through an NIH T32 grant (T32 CA080416) and ETG reviewed funding from the DoD BCRP Breakthrough Fellowship Award (W81XWH-19-1-0076). This research was also supported by P01 CA163227 and R01 CA234715 to PSN, the Prostate Cancer Foundation, the Genomics and Bioinformatics Shared Resource of the Fred Hutch/University of Washington Cancer Consortium (P30 CA015704) and the Scientific Computing Infrastructure at Fred Hutch funded by ORIP grant S10 OD028685.

## Additional information

### Funding

| Funder | Grant reference number | Author |
| --- | --- | --- |
| National Cancer Institute | R37 CA230617 | Andrew C Hsieh |
| National Institute of General Medical Sciences | R01 GM135362 | Andrew C Hsieh |
| National Cancer Institute | P50 CA097186 | Peter S Nelson |
| Burroughs Wellcome Fund | 1012314.02 | Andrew C Hsieh |
| Robert J. Kleberg, Jr. and Helen C. Kleberg Foundation | | Andrew C Hsieh |
| National Cancer Institute | T32 CA080416 | Alexandre A Germanos |
| Department of Defense | BCRP W81XWH-19-1-0076 | Erica T Goddard |
| National Cancer Institute | P01 CA163227 | Peter S Nelson |
| National Cancer Institute | R01 CA234715 | Peter S Nelson |
| Prostate Cancer Foundation | | Peter S Nelson |
| National Cancer Institute | P30 CA015704 | Andrew C Hsieh |
| National Cancer Institute | S10 OD028685 | Andrew C Hsieh |
| Emerson Collective | 691630 | Andrew C Hsieh |

The funders had no role in study design, data collection and interpretation, or the decision to submit the work for publication.

### Author contributions

Alexandre A Germanos, Conceptualization, Data curation, Formal analysis, Funding acquisition, Investigation, Methodology, Validation, Visualization, Writing – original draft, Writing – review and editing; Sonali Arora, Data curation, Formal analysis, Methodology, Software, Visualization, Writing – review and editing; Ye Zheng, Formal analysis, Methodology; Erica T Goddard, Cyrus M Ghajar, Formal analysis, Validation; Ilsa M Coleman, Formal analysis, Visualization; Anson T Ku, Formal analysis, Methodology, Visualization; Scott Wilkinson, Annalysa Long, Jason H Bielas, Validation, Writing – review

and editing; Hanbing Song, Validation, Formal analysis; Nicholas J Brady, Data curation, Formal analysis, Methodology, Visualization; Robert A Amezquita, Validation, Methodology, Software; Michael Zager, Data curation, Supervision, Visualization; Yu Chi Yang, David S Rickman, Methodology, Supervision; Raphael Gottardo, Methodology, Software; Franklin W Huang, Methodology, Validation; Peter S Nelson, Validation, Methodology; Adam G Sowalsky, Data curation, Formal analysis, Supervision; Manu Setty, Formal analysis, Methodology, Software, Visualization; Andrew C Hsieh, Conceptualization, Funding acquisition, Supervision, Writing – review and editing

Author ORCIDs
Alexandre A Germanos http://orcid.org/0000-0003-4677-1313
Michael Zager http://orcid.org/0000-0002-9416-8685
Franklin W Huang http://orcid.org/0000-0001-5447-0436
Adam G Sowalsky http://orcid.org/0000-0003-2760-1853
Andrew C Hsieh http://orcid.org/0000-0002-0897-1050

Ethics
All mice were maintained in the C57BL/6 background under specific pathogen-free conditions, and experiments conformed to the guidelines as approved by the Institutional Animal Care and Use Committee (#50870) of Fred Hutchinson Cancer Research Center (FHCRC).

Decision letter and Author response
Decision letter https://doi.org/10.7554/eLife.79076.sa1
Author response https://doi.org/10.7554/eLife.79076.sa2

# Additional files

Supplementary files
• Supplementary file 1. Quality control metrics and *Figure 1* supplementary information. (A) Quality control measures for each mouse replicate. (B) Breakdown of cell ID numbers and relative abundance (%) per mouse condition. (C) Breakdown of cell ID numbers and relative abundance (%) per mouse replicate. (D) Genes upregulated in *Pten*$^{fl/fl}$ compared to WT mice for each epithelial subtype. Thresholds set at avg_log2FC >0.25 and FDR <0.05. (E) Breakdown of cell cycle phase assignment for every cell ID in WT and *Pten*$^{fl/fl}$ mice. (F) Differentially expressed genes between hyper-proliferative and hypo-proliferative basal cells in *Pten*$^{fl/fl}$ mice. Thresholds set at avg_log2FC >0.25 and FDR <0.05. Positive avg_log2FC values indicate upregulation in hypo-proliferative basal cells.

• Supplementary file 2. CellphoneDB cell-cell interaction data in *Pten*$^{fl/fl}$ mice (see *Figure 2*). The first gene in the "pair" column is expressed in the first cell ID in the "clusters" column and the second gene is expressed in the second cell ID.

• Supplementary file 3. Differentially expressed genes in *Pten*$^{fl/fl}$ and *Pten*$^{fl/fl}$ cx intermediate cells. 6-way comparison between clusters 0–5 (see *Figure 3E*). Each sheet shows significantly up- or down-regulated genes for one cluster relative to all others. Thresholds set at avg_log2FC >0.25 and FDR <0.05.

• Supplementary file 4. CellphoneDB cell-cell interaction data in *Pten*$^{fl/fl}$ cx mice (see *Figure 5*). The first gene in the "pair" column is expressed in the first cell ID in the "clusters" column and the second gene is expressed in the second cell ID.

• Supplementary file 5. Differentially expressed genes and CellphoneDB cell-cell interaction data in *Pten*$^{fl/fl}$;4ebp1$^{M}$ cx mice (see *Figure 6*). (A) Genes upregulated in *Pten*$^{fl/fl}$;4ebp1$^{M}$ cx compared to *Pten*$^{fl/fl}$ cx mice for each epithelial subtype. Thresholds set at avg_log2FC >0.25 and FDR <0.05. (B) Genes upregulated in *Pten*$^{fl/fl}$;4ebp1$^{M}$ cx compared to *Pten*$^{fl/fl}$ cx mice for each epithelial subtype, filtered for cells expressing the *rtTA-eGFP* transgene. Thresholds set at avg_log2FC >0.25 and FDR <0.05. (C) CellphoneDB cell-cell interaction data in *Pten*$^{fl/fl}$;4ebp1$^{M}$ cx mice (see *Figure 6E–H*). The first gene in the "pair" column is expressed in the first cell ID in the "clusters" column and the second gene is expressed in the second cell ID.

• MDAR checklist

## Data availability

Sequencing data have been deposited in GEO under accession code GSE17133. All code used for this study is available on Github at https://github.com/sonali-bioc/GermanosProstatescRNASeq/ (copy archived at swh:1:rev:5a376d7b77d034e9bd09ce4787337ee33fda8448).

The following dataset was generated:

| Author(s) | Year | Dataset title | Dataset URL | Database and Identifier |
|-----------|------|---------------|-------------|------------------------|
| Germanos A, Arora S, Hsieh A | 2022 | Defining Cellular Population Dynamics in Advanced Prostate Cancer using Single-cell RNA Sequencing | https://www.ncbi.nlm.nih.gov/geo/query/acc.cgi?acc=GSE171336 | NCBI Gene Expression Omnibus, GSE171336 |

The following previously published datasets were used:

| Author(s) | Year | Dataset title | Dataset URL | Database and Identifier |
|-----------|------|---------------|-------------|------------------------|
| Network TCGAR | 2015 | TCGA | https://portal.gdc.cancer.gov/ | GDC Data Portal, TCGA |
| Taylor BS, Schultz N, Hieronymus H, Gopalan A, Xiao Y, Carver BS, Arora Vivek K, Major John E, Socci Nicholas D, Lash Alex E, Eastham James A, Reuter Victor E, Scardino Peter T, Sawyers Charles L, Gerald WL, Poorvi K, Ethan C, Boris R, Yevgeniy A, Nicholas M, Thomas L, Igor D, Manda W, Adriana H, Howard S, Chris S | 2010 | Prostate Adenocarcinoma (MSK, Cancer Cell 2010) | http://www.cbioportal.org/study/summary?id=prad_mskcc | cBioPortal, prad_mskcc |
| Dong B, Miao J, Wang Y, Luo W, Ji Z, Lai H, Zhang M, Cheng X, Jinming W, Fang Y, Zhu HH, Chua CW, Fan L, Zhu Y, Pan J, Jia W, Xue W, Gao WQ | 2020 | Single-cell analysis supports a luminal-neuroendocrine transdifferentiation in human prostate cancer | https://www.ncbi.nlm.nih.gov/geo/query/acc.cgi?acc=GSE137829 | NCBI Gene Expression Omnibus, GSE137829 |
| Song H, Weinstein HNW, Allegakoen P, Wadsworth MH, Xie J, Yang H, Castro EA, Stohr BA, Feng FY, Carroll PR, Wang B, Cooperberg MR, Shalek AK, Huang FW | 2022 | Single-cell analysis of human primary prostate cancer reveals the heterogeneity of tumor-associated epithelial cell states | https://www.ncbi.nlm.nih.gov/geo/query/acc.cgi?acc=GSE176031 | NCBI Gene Expression Omnibus, GSE176031 |
| Brady NJ, Bagadion AM, Singh R, Conteduca V, Emmenis LV, Arceci E, Pakula H, Carelli R, Khani F, Bakht M, Sigouros M, Bareja R, Sboner A, Elemento O, Tagawa S, Nanus DM, Loda M, Beltran H, Robinson B, Rickman DS | 2021 | Temporal evolution of cellular heterogeneity during the progression to advanced AR-negative prostate cancer | https://www.ncbi.nlm.nih.gov/geo/query/acc.cgi?acc=GSE151426 | NCBI Gene Expression Omnibus, GSE151426 |

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
