## [Editor Report]

Prostate cancer cellular heterogeneity is a major problem for disease progression and treatment resistance. This body of work addresses the cellular identity and populations that make up prostate cancer using single-cell sequencing technology, state-of-the-art mouse models and connections to human prostate cancer. The cellular identities, associated signaling networks, and immune complexes accompanying the heterogeneity of the prostate are identified in this work and a resource is provided for scientists in the field.

---

## [Decision Letter]

**Decision letter after peer review:**

Thank you for submitting your article "Defining cellular population dynamics at single cell resolution during prostate cancer progression" for consideration by *eLife*. Your article has been reviewed by 2 peer reviewers, and the evaluation has been overseen by a Reviewing Editor and W Kimryn Rathmell as the Senior Editor. The following individual involved in the review of your submission has agreed to reveal their identity: Douglas Strand (Reviewer #2).

Essential revisions:

This manuscript has significant strengths that could make it an asset to the field. However, there are also deficiencies that need to be addressed and require major revisions. Based on the referee evaluations, the following are the major points that need to be addressed:

1) The 'intermediate' cell state observed in the intact and castrated PTEN model is not observed in primary prostate cancer and there is little sc evidence yet in advanced cancer to compare it to.

2) There should be some additional emphasis placed on the human level validation for the central findings in this work.

3) Please address the deficiencies of using the PB-Cre-PTEN model as detailed by the reviewers.

4) Please address the limitations of using trajectory analysis. Please see reviewer 2 comments.

*Reviewer #1 (Recommendations for the authors):*

The study focuses on a clinically highly relevant and timely topic. The strength of this manuscript is the meticulous description of the Methods and model development and the integration of state-of-the-art orthogonal data sets. However, the number of data-points across the experiments (n = 2 or 3) with considerable variability in the Ptenfl/fl group limits the interpretation of findings. Additionally, further experiments are needed to validate these observations in a human prostate cancer and establish potential translational relevance of these findings. As such, the report is fairly descriptive and expanding the discussion on the mechanistic studies needed to identify which of these interactions drives aggressive prostate cancer would improve this report.

Pg 6, lines 254-256. Where is the role of epithelial cells discussed and supported by findings?

Pg 7, lines 279-280. I'm not sure that "early on in cancer" is supported by the data. Experiments focused on signaling patterns and cell-to-cell interactions, but time course of disease isn't shown/emphasized until the conclusion statement.

Pg 7, lines 290-304. AR activity in intermediate cells was compared between WT, intact and castrated PTEN-null samples. Line 300, differentiated cells retain high AR activity in intact PTEN-null but because population is absent in castrate mice (line 296), how is final conclusion about "luminal cells", broadly, arrived at?

Pg 8, lines 339-363. Were the TCGA cohorts confirmed PTEN loss? Please clarify and mention limitations when comparing DEGs in mice with human cohorts.

*Reviewer #2 (Recommendations for the authors):*

Unfortunately, the expansion of an intermediate cell state in Pten null mice (PMID: 28603917) or castrated mice (PMID: 32355025) is not very new and diminishes the impact. Also, the stem cell story has evolved quite a bit since 2016. 'Intermediate luminal cells' in the proximal prostate are preexisting Krt4+ urethral cells (PMID: 32497356). This is only a 'rare, but distinct population' in the prostate. In contrast, the entire urethra is also composed of these cells. Lineage tracing of these proximal cells during castration/regeneration shows that they do NOT give rise to prostate luminal cells even though they form spheroids at a higher rate (PMID: 32355025, 32807988). Instead, preexisting prostate luminal cells undergo lineage plasticity to resemble urethral/intermediate luminal cells (yes, it's confusing – why would a cell adapt to resemble a cell type that isn't a progenitor for it? – I don't know). Accordingly, it is inaccurate to say that these preexisting cells 'fill an important regenerative niche' until there is opposing evidence. We all know that RNA velocity can make anything look like a progenitor. Also, the urothelium of the bladder also expresses Psca, Trop2, and Krt4 and can regenerate prostate tissue in tissue recombination assays – this does not mean that bladder urothelium is a prostate progenitor.

The use of the term 'intermediate' luminal as a descriptive term for the plastic state of prostate luminal cells during castration, cancer is just ok. However, this must be distinguished from 'urethral' luminal epithelia, which are a pre-existing proximal cell type in the urethra and proximal ducts of the intact prostate that does not express NKX3-1 or probasin. The key is to figure out how/why prostate luminal cells adapt to look like (and cluster with) urothelium under pten inactivation or castration (perhaps KL5? PMID: 34737261). Unfortunately, I think the leaky inactivation of Pten in basal and urethral luminal cells is confounding this question in this model. Accordingly, is there any solid evidence yet of basal epithelia acquiring mutations in human prostate cancer (notwithstanding the experimental evidence from Goldstein)?

What would be interesting is to highlight the molecular difference between lineage plastic prostate luminal cells in the castrate WT and intact Pten null mice and compare to the intact WT proximal urethral cell signature. This would give us an idea of how similar the lineage plastic state is under the 2 different conditions (I agree with your suggestion that pten inactivation could simply reduce Ar activity, making it undergo similar lineage plasticity to an 'intermediate' state). In summary, make sure to distinguish between the preexisting proximal urethral luminal cells that don't give rise to prostate luminal cells during castration/regeneration and the 'intermediate' prostate luminal cells that underwent plasticity under castration and Pten inactivation and can likely regain prostate differentiation when androgens are replenished.

Figure 1: The proposition that basal cells are differentiating into intermediate luminal cells is intriguing, but could just as easily be an artifact of trajectory analysis (which is notoriously misleading) or the activation of Pten in basal epithelia (which is not yet known to happen in human prostate cancer as far as I know). It also defies the current paradigm that prostate luminal epithelia undergo lineage plasticity in castration and when mutated. Therefore, I would suggest hedging this bet in the conclusion.

Figure 3: While intermediate/urethral epithelia may be AR+ in WT mice, their expression of probasin and NKX3-1 is null. Thus, the AR composite score is misleading in Figure 3D. Moreover, how would the AR score be higher in intermediate/urethral epithelia than in differentiated luminal cells in WT mice??? It would seem the Hieronymus score from LNCaP cells is inappropriately used here as a measure of ALL AR activity and not a measure of prostate luminal-specific AR activity. A more accurate score would be to use probasin expression itself or a signature generated from WT vs. castrated mice (PMID: 32355025).

The color coding of the subclustered intermediate epithelia in 3E does not seem to match the quantitation in 3F. This confuses the interpretation of 3G as well. It seems likely that the heterogeneity of cells identified as intermediate epithelia could reflect cells that are transdifferentiating from basal due to promiscuous Pten inactivation as well as from lineage plasticity of Pten inactivation in prostate luminal cells. It would be really useful to see how these clusters compare to the castrate ventral lobe in a WT mouse. It would also be useful to see the AR activity of these clusters relative to intact prostate basal and luminal cells in Figure 3G.

I'm not sure that the observation that castration increases the number of AR-low intermediate cells necessarily means that this is a selection of a pre-existing state since we know that pre-existing prostate luminal epithelia can undergo lineage plasticity in response to castration in the non-tumorigenic setting.

The observation that the castrate Pten intermediate epithelial signature is enriched in aggressive disease (compared to intact Pten intermediate) is not too surprising as this may just reflect castration. What would be interesting is to know how this model compares to other models of castration resistance like Rb/p53 null mice.

Discussion:

Line 488: While it's true that the transdifferentiation, an intermediate luminal phenotype is absent in the (intact) wild-type mouse, this phenotype does appear in castrate WT mice. This is not a cancer-specific lineage transition as this also happens in 5ARI-treated men (PMID 34928497).

Line 499: I'm not convinced that these data suggest that specific portions of the intermediate compartment are responsible for their expansion. Without lineage tracing in humans, it's impossible to know whether the change in heterogeneity is the result of increased transdifferentiation of both basal and luminal cells. This may be artificially induced by the promiscuous PB4-Cre promoter rather than a reflection of human prostate cancer as the level of CNVs in basal epithelia in primary tumors appears to be low.

---

## [Author Response]

Essential revisions:This manuscript has significant strengths that could make it an asset to the field. However, there are also deficiencies that need to be addressed and require major revisions. Based on the referee evaluations, the following are the major points that need to be addressed:1) The 'intermediate' cell state observed in the intact and castrated PTEN model is not observed in primary prostate cancer and there is little sc evidence yet in advanced cancer to compare it to.

We thank the reviewers for pointing out a potential pitfall of our study. In response, we have now analyzed human patient-based scRNAseq data from localized (Song et al., Nat Comms 2022) and metastatic prostate cancer (Dong et al., Comms Bio 2020). We found that the gene signatures derived from our intermediate cells are enriched in the setting of castration resistant prostate cancer (Figure 4D-E). We conclude that an intermediate-like state likely exists in advanced human prostate cancer and potentially contributes to patient outcomes.

2) There should be some additional emphasis placed on the human level validation for the central findings in this work.

We agree with the reviewers that further confirmation that our findings apply to human disease would strengthen this manuscript. As such, we have added single-cell RNA-seq data of human tumors to validate our observations. We used data from multiple human studies (Song et al., Nat Comms 2022; Dong et al., Comms Bio 2020) to compare primary and metastatic prostate cancer. We show that our 5-gene treatment resistance signature from our murine dataset correlates with a subset of cells from 6 human metastatic prostate cancers but is completely absent in localized disease (Figure 4D-E). We conclude that castrated intermediate cells correlate to a cell state found in human metastatic castration resistant prostate cancer tumors, and that this subset may contribute to ADT resistance.

3) Please address the deficiencies of using the PB-Cre-PTEN model as detailed by the reviewers.

We thank the reviewers for pointing this out. To address the deficiencies of using the *PB-Cre-Pten* model, we have now conducted additional analyses across orthogonal models and patient cohorts. In particular, we have validated our key findings in another series of mouse models of castration-resistant prostate cancer (Brady et al., Nat Comms 2021) and have used public patient datasets (TCGA, Taylor) as well as single-cell sequencing data from primary and metastatic prostate cancer patients (Song et al., Nat Comms 2022; Dong et al., Comms Bio 2020) to correlate our observations to clinically relevant models (Figure 4D-G). In addition, we discuss that the PB-Cre driver is active in multiple epithelial subtypes in the *Pten^fl/fl^* model, and therefore new lineage tracing studies will be required to determine the origins of the intermediate cell states (Lines 617-629).

4) Please address the limitations of using trajectory analysis. Please see reviewer 2 comments.

We agree with the reviewers that trajectory analysis alone is not sufficient to establish cellular lineages and phenotype switches. We have updated our analysis of our data and integrated further literature to speculate about the cellular origins of intermediate cells in the *Pten* loss model (Lines 262-276), and believe our hypotheses are now in line with the state of the field. We have edited the text to clarify the need for in-tissue validation of our computational findings (Lines 274-276, 626-629).

Reviewer #1 (Recommendations for the authors):The study focuses on a clinically highly relevant and timely topic. The strength of this manuscript is the meticulous description of the Methods and model development and the integration of state-of-the-art orthogonal data sets. However, the number of data-points across the experiments (n = 2 or 3) with considerable variability in the Ptenfl/fl group limits the interpretation of findings. Additionally, further experiments are needed to validate these observations in a human prostate cancer and establish potential translational relevance of these findings. As such, the report is fairly descriptive and expanding the discussion on the mechanistic studies needed to identify which of these interactions drives aggressive prostate cancer would improve this report.

We are ecstatic that the reviewer finds our study “clinically highly relevant.” We agree that the low sample size is a potential limitation but believe that our overall results are robust and enable concrete conclusions for both epithelial and immune cell populations. This is in part because we validated our findings in orthogonal human datasets (Figure 4A-C, Figure 5H) in the original manuscript. However, to add rigor to our study, we have conducted new scRNAseq analysis showing that our findings correlate well with both human patient data (Figure 4D-E) and orthogonal mouse models (Figure 4F-G). Furthermore, we conducted additional scRNAseq on castrated WT murine prostate to demonstrate how castration plays an important role in translational heterogeneity in intermediate cells (Figure 4H, Figure 3–figure supplement 1G).

As such, the report is fairly descriptive, and expanding the discussion on the mechanistic studies needed to identify which of these interactions drives aggressive prostate cancer would improve this report.

We agree with the reviewer that additional discussion of follow-up studies is necessary. As such, we have updated the discussion to highlight the molecular studies needed to fully characterize the cellular phenotypes described in this manuscript (Lines 626-629, 643-644, 658- 660, 676-678).

Pg 6, lines 254-256. Where is the role of epithelial cells discussed and supported by findings?

We outline epithelial-macrophage signaling interactions in lines 310-315 (highlighted in the text), showing both the presence of ligand-receptor interactions and the increased expression of epithelial ligands in Ptenfl/fl mice.

Pg 7, lines 279-280. I'm not sure that "early on in cancer" is supported by the data. Experiments focused on signaling patterns and cell-to-cell interactions, but time course of disease isn't shown/emphasized until the conclusion statement.

We thank the reviewer for pointing out this passage, as we agree that it is impossible to infer accurate temporal dynamics from our data. We have rephrased the passage to more precisely reflect the nature of our observations (Lines 350-351).

Pg 7, lines 290-304. AR activity in intermediate cells was compared between WT, intact and castrated PTEN-null samples. Line 300, differentiated cells retain high AR activity in intact PTEN-null but because population is absent in castrate mice (line 296), how is final conclusion about "luminal cells", broadly, arrived at?

We thank the reviewer for pointing out confusing language in the text. We have updated both Figure 3D and the text to better reflect our hypothesis. Specifically, we find that while intermediate cells exhibit a lower reliance on AR signaling upon Pten loss, differentiated luminal cells do not. We have also added speculation that the ablation of differentiated luminal cells in Ptenfl/fl castrate mice could be due to lineage plasticity rather than apoptosis in response to AR deprivation (Lines 374-378).

Pg 8, lines 339-363. Were the TCGA cohorts confirmed PTEN loss? Please clarify and mention limitations when comparing DEGs in mice with human cohorts.

The original plot was not filtered by PTEN status. We agree with the reviewer that stratifying the data for PTEN loss would be valuable and reanalyzed the data with this in mind. We find that filtering patients for either PTEN loss or amplification of the PI3K/AKT pathway in the TCGA data yields similar results, where patients overexpressing any of the 5 genes of interest have significantly lowered disease-free survival (Figure 4 – figure supplement 1D). The Taylor dataset did not have a large enough sample size for this stratification strategy to yield robust data.

We also recognize that comparing specific DEGs across species can lead to artifacts. As such, we have now validated our 5-gene signature in two additional systems. First, using scRNAseq data from localized (Song et al., Nat Comms 2022) and metastatic (Dong et al., Comms Bio 2020) prostate cancer patients, we found that only the latter had cancer cells enriched for the 5- gene signature (Figure 4D-E). Second, using a completely different progression model of murine prostate cancer (Brady et al., Nat Comms 2021), we found that the 5-gene signature correlated with advanced murine prostate cancer (Figure 4F-G). These data demonstrate the robustness of our original findings and overcome some of the limitations when comparing DEGs across mice and men.

Reviewer #2 (Recommendations for the authors):Unfortunately, the expansion of an intermediate cell state in Pten null mice (PMID: 28603917) or castrated mice (PMID: 32355025) is not very new and diminishes the impact.

We agree with the reviewer that the presence of intermediate cells in the *Pten^fl/fl^* mouse model is not novel. However, our findings go far beyond just identifying an intermediate population that expands during castration. In particular, we show that intermediate cells exhibit increased heterogeneity in the context of castration and may originate from multiple cellular sources in the *Pten^fl/fl^* model. We also show that the castrate *Pten^fl/fl^* intermediate cell state is associated with human prostate cancer metastasis (Figure 4D-E), worse outcomes in prostate cancer patients (Figure 4A-C, Figure 4 —figure supplement 1C-D), and can be seen in an entirely different murine model of advanced prostate cancer (Figure 4F-G). Thus, our work adds to the growing understanding of *Pten^fl/fl^* intermediate cell dynamics and its relevance to human disease.

Also, the stem cell story has evolved quite a bit since 2016. 'Intermediate luminal cells' in the proximal prostate are preexisting Krt4+ urethral cells (PMID: 32497356). This is only a 'rare, but distinct population' in the prostate. In contrast, the entire urethra is also composed of these cells. Lineage tracing of these proximal cells during castration/regeneration shows that they do NOT give rise to prostate luminal cells even though they form spheroids at a higher rate (PMID: 32355025, 32807988). Instead, preexisting prostate luminal cells undergo lineage plasticity to resemble urethral/intermediate luminal cells (yes, it's confusing – why would a cell adapt to resemble a cell type that isn't a progenitor for it? – I don't know). Accordingly, it is inaccurate to say that these preexisting cells 'fill an important regenerative niche' until there is opposing evidence. We all know that RNA velocity can make anything look like a progenitor. Also, the urothelium of the bladder also expresses Psca, Trop2, and Krt4 and can regenerate prostate tissue in tissue recombination assays – this does not mean that bladder urothelium is a prostate progenitor.

We thank the reviewer for this detailed response and agree on the need to distinguish between urethral progenitor cells and native prostatic cells. We had previously excluded a cluster of cells from our dataset that we had identified as urethral cells; we re-included this cluster and confirmed that it primarily exists in WT intact mice and expresses urethral markers such as *Psca* and *Krt4* (Figure 1 —figure supplement 1C-D). We also compared this cluster to the previously described “intermediate” cluster in the WT intact mice and showed that the new urethral cluster, unlike the “intermediate” WT cluster, expresses the urethral markers *Aqp3* and *Ly6d* but not luminal progenitor markers *Ppp1r1b* and *Clu* (Figure 1 —figure supplement 1C), and do not cluster with WT “intermediate” cells (Figure 1 —figure supplement 1D). This matches well with published data identifying urethral and prostatic *Krt4+* cells in murine proximal prostates (Crowley et al., *eLife* 2020). In addition, the urethral cluster exhibits weaker expression of prostate epithelial and luminal markers (*Epcam*/*Krt18*/*Krt8*) (Figure 1 —figure supplement 1H). These results indicate that while the excluded cluster is likely urethral in nature, the previously included WT “intermediate” cluster, which is *Epcam/Krt18/Krt8* positive, may be composed of prostatic cells. Based on this analysis, we believe the previously named WT “intermediate” cluster may correspond to the proximal *Krt4+* cells described in Crowley et al. (*eLife* 2020) and the distal progenitor cells found in Guo et al. (Nat Genetics 2020). As such, we have re-named the WT “intermediate” cells to “luminal progenitor” cells (Figure 1 —figure supplement 1D). Moreover, given that urethral cells do not give rise to prostatic cells and are thus uninvolved in prostate cellular dynamics, we excluded this cluster from further analysis.

The use of the term 'intermediate' luminal as a descriptive term for the plastic state of prostate luminal cells during castration, cancer is just ok. However, this must be distinguished from 'urethral' luminal epithelia, which are a pre-existing proximal cell type in the urethra and proximal ducts of the intact prostate that does not express NKX3-1 or probasin.

As outlined above, we have identified urethral cells in our dataset. These cells cluster apart from both WT luminal progenitor and *Pten^fl/fl^* intermediate cells and express urethral-specific markers (Figure 1 —figure supplement 1C-H). We have also excluded these cells from our analysis (Lines 192-193).

The key is to figure out how/why prostate luminal cells adapt to look like (and cluster with) urothelium under pten inactivation or castration (perhaps KL5? PMID: 34737261). Unfortunately, I think the leaky inactivation of Pten in basal and urethral luminal cells is confounding this question in this model. Accordingly, is there any solid evidence yet of basal epithelia acquiring mutations in human prostate cancer (notwithstanding the experimental evidence from Goldstein)?

We agree with the reviewer that the ubiquitous inactivation of *Pten* in multiple prostatic epithelial cell types in the *Pten^fl/fl^* model makes it very difficult to determine how/why prostate luminal cell adapt to looks like urothelium. In order to study this, precise lineage tracing and subsequent mechanistic studies using cell-type specific models are needed to begin to understand this very interesting and confounding biology. We have now discussed these technologies in Lines 626-629 and 643-644.

As the reviewer has mentioned, human prostatic basal cells can transform into adenocarcinoma as shown by the Goldstein group using ex vivo models (Stoyanova et al., PNAS 2013). However, definitive evidence that this occurs in human tissue is currently lacking. We analyzed the basal cells in the Song et al. (Nat Comms 2022) scRNAseq dataset and found no indication of prostate cancer specific genetic lesions. However, this analysis was limited by a small sample size of n = 11. What we do know about prostatic basal cell transformation in humans can be found in case reports. It has been shown that 0.01% of human prostate cancer can present with low PSA and basal cell markers such as p63, CK5/6, and 34bE12 (Ninomiya et al., Case Reports in Oncology 2018; Dong et al., Frontiers in Oncology 2020; Shibuya et al., Molecular and Clinical Oncology 2018; Kun et al., World J Surg Oncol 2013). This is the best evidence that basal cell transformation does occur in humans. Whether transdifferentiation ensues after transformation remains to be determined.

What would be interesting is to highlight the molecular difference between lineage plastic prostate luminal cells in the castrate WT and intact Pten null mice and compare to the intact WT proximal urethral cell signature. This would give us an idea of how similar the lineage plastic state is under the 2 different conditions (I agree with your suggestion that pten inactivation could simply reduce Ar activity, making it undergo similar lineage plasticity to an 'intermediate' state). In summary, make sure to distinguish between the preexisting proximal urethral luminal cells that don't give rise to prostate luminal cells during castration/regeneration and the 'intermediate' prostate luminal cells that underwent plasticity under castration and Pten inactivation and can likely regain prostate differentiation when androgens are replenished.

We agree with the reviewer’s suggestion of using WT castrate mice as a comparison point with *Pten^fll/fl^* intact mice. To address this question, we conducted new scRNAseq on 9 WT castrated mice, which we binned into 3 replicates, and integrated them into our existing dataset. We find that non-basal cells in WT castrate mice aggregate into one clustered region separate from both WT intact luminal cells and *Pten^fl/fl^* intermediate cells (Figure 3 —figure supplement 1G). Interestingly, WT castrate (cx) luminal cells cluster closest to WT luminal progenitor cells (Figure 3 —figure supplement 1G). Computing signature scores for AR activity, translation, and proliferation between these 3 cell types reveals that WT castrate luminal cells are most similar to AR-low *Pten^fl/fl^* intermediate cells, while WT intact cells share features with AR-high *Pten^fl/fl^* intermediate cells (Figure 3G-H). We conclude that “intermediate-like” cells can exhibit multiple distinct states partly depending on AR activity, *Pten* loss, and castration status.

Figure 1: The proposition that basal cells are differentiating into intermediate luminal cells is intriguing, but could just as easily be an artifact of trajectory analysis (which is notoriously misleading) or the activation of Pten in basal epithelia (which is not yet known to happen in human prostate cancer as far as I know). It also defies the current paradigm that prostate luminal epithelia undergo lineage plasticity in castration and when mutated. Therefore, I would suggest hedging this bet in the conclusion.

We thank the reviewer for this suggestion. Through additional analysis, we show that intermediate cells in *Pten^fl/fl^* mice represent a heterogeneous population with multiple potential cellular origins, including basal, luminal progenitor, and luminal prostate epithelial cells. In addition to our findings via trajectory analysis, RNA velocity, and Palantir (Figure 1H-J, Figure 1 —figure supplement 2C), we find that a subset of *Pten^fl/fl^* intermediate cells express basal markers (Figure 3 —figure supplement 1B), supporting the hypothesis that some intermediate cells are derived from basal cell transdifferentiation. In addition, we have generated new evidence of luminal phenotype switching in our dataset. Specifically, we observe expression of multiple luminal markers in *Pten^fl/fl^* intermediate cells immediately adjacent to differentiated luminal cells, suggesting that luminal cells undergo lineage plasticity to resemble intermediate cells while maintaining some identifying markers upon tumorigenesis (Figure 1 —figure supplement 2E). Finally, we show that WT intact mice contain both urethral and prostatic progenitor cells (Figure 1 —figure supplement 1C-H). Given that both proximal and distal *Krt4+* cells have been shown to be tumor initiating in the context of *Pten* loss (Guo et al., Nat Genetics 2020), it is possible that a subset of intermediate cells originated from these WT luminal progenitor cells. Given these 3 possible origins, we have updated our conclusions to reflect the multiple possible cellular origins of intermediate cell expansion in *Pten^fl/fl^* mice (Lines 257-276, and 617-629).

Figure 3: While intermediate/urethral epithelia may be AR+ in WT mice, their expression of probasin and NKX3-1 is null. Thus, the AR composite score is misleading in Figure 3D. Moreover, how would the AR score be higher in intermediate/urethral epithelia than in differentiated luminal cells in WT mice??? It would seem the Hieronymus score from LNCaP cells is inappropriately used here as a measure of ALL AR activity and not a measure of prostate luminal-specific AR activity. A more accurate score would be to use probasin expression itself or a signature generated from WT vs. castrated mice (PMID: 32355025).

We thank the reviewer for bringing up this concern. As outlined above, we find that WT “intermediate” cells are likely not urethral in origin, and we reclassify them separately from intermediate cells as luminal progenitor cells. We have re-generated Figure 3D to include the distinction between WT luminal progenitor and *Pten^fl/fl^* intermediate cells and observe that luminal progenitor cells exhibit AR activity, while total AR activity in *Pten^fl/fl^* intermediate cells is low regardless of castration context (Figure 3D). This is consistent with previous studies that show unlike urethral cells, prostatic *Krt4+* cells express AR (Guo et al., Nat Genetics 2020; Crowley et al., *eLife* 2020).

The color coding of the subclustered intermediate epithelia in 3E does not seem to match the quantitation in 3F. This confuses the interpretation of 3G as well. It seems likely that the heterogeneity of cells identified as intermediate epithelia could reflect cells that are transdifferentiating from basal due to promiscuous Pten inactivation as well as from lineage plasticity of Pten inactivation in prostate luminal cells. It would be really useful to see how these clusters compare to the castrate ventral lobe in a WT mouse. It would also be useful to see the AR activity of these clusters relative to intact prostate basal and luminal cells in Figure 3G.

We have corrected the misaligned colors in Figure 3E-F. We agree with the reviewer that multiple cells of origin could give rise to the highly heterogeneous intermediate compartment. To investigate how the changes in intermediate cells between *Pten^fl/fl^* intact and castrated mice observed in Figure 3E-G compare to a WT castrate prostate, we conducted scRNAseq of 9 castrated WT mouse prostates. The new Figure 3H shows AR activity, proliferation, and translation scores for basal and luminal cells in WT intact and castrate mice. We observe that in castration, basal cells lose proliferative capacity, while WT luminal cells lose AR activity but exhibit increased expression of translation machinery genes, most closely resembling AR-low intermediate cells in the *Pten^fl/fl^* mice (Figure 3G-H). As these AR-low intermediate cells expand in *Pten^fl/fl^* mice upon castration, we conclude that translational hyperactivation is likely a direct consequence of castration in the mouse prostate, which corresponds well to the observation that castration leads to increased mRNA translation in the *Pten^fl/fl^* model (Liu et al., Sci Trans Med 2019). However, contrary to *Pten^fl/fl^* AR-low intermediate cells, we did not observe a concomitant increase in proliferation score in WT castrate luminal cells. These findings demonstrate that castration plays an important role in translational heterogeneity in *Pten^fl/fl^* intermediate cells. However, the change in proliferation is likely related to loss of *Pten.*

I'm not sure that the observation that castration increases the number of AR-low intermediate cells necessarily means that this is a selection of a pre-existing state since we know that pre-existing prostate luminal epithelia can undergo lineage plasticity in response to castration in the non-tumorigenic setting.

We thank the reviewer for the intriguing suggestion that AR-low intermediate cells may represent an additional phenotype switching event. We have confirmed that AR-low, translation-high, proliferation-high cells exist in low numbers in the intact *Pten^fl/fl^* mice (Figure 3F – top panel, Figure 3G). Therefore, the expansion of these cells in castrate mice is the most direct explanation for their presence. However, we recognize it is also possible that castration provokes new lineage plasticity events. Absent lineage tracing studies in vivo we cannot make conclusive statements about the precise cellular origins of hyperproliferative intermediate cells. We have updated the results and Discussion section (Lines 415-425, 640-644) to reflect this uncertainty.

The observation that the castrate Pten intermediate epithelial signature is enriched in aggressive disease (compared to intact Pten intermediate) is not too surprising as this may just reflect castration. What would be interesting is to know how this model compares to other models of castration resistance like Rb/p53 null mice.

We agree with the reviewer that validation of the prognostic signature in another model of castration resistance would be valuable. We have initiated a collaboration with the Rickman lab to correlate our findings with orthogonal mouse models of cancer progression and castration resistance. Specifically, we have projected our 5-gene signature of treatment resistance onto data from Brady et al. (Nat Comms 2021) using their double- and triple-mutant mice (*Pten^-/-^/Rb1^-/-^* and *Pten^-/-^/Rb1^-/-^/Nmyc^+^*). The triple mutant (PRN) mice exhibit much stronger castration resistance than the double mutant (PR). Single-cell RNA sequencing data in this study was produced without castration in both PR and PRN mice, resulting in a model of cancer progression outside of physical castration. We show that our 5-gene signature is much more strongly expressed in the PRN mice compared to the PR mice, validating that the signature is enriched in a model of castration resistance (Figure 4F-G).

Discussion:Line 488: While it's true that the transdifferentiation, an intermediate luminal phenotype is absent in the (intact) wild-type mouse, this phenotype does appear in castrate WT mice. This is not a cancer-specific lineage transition as this also happens in 5ARI-treated men (PMID 34928497).

We do note a shift of luminal cells in WT castrate mice towards luminal progenitors and intermediate cells, although they cluster separately from both cell states as well as proximal urethral cells (Figure 3 —figure supplement 1G). The WT castrate luminal cells cluster closest to WT intact luminal progenitor cells, indicating they may be closely related to this cell type (Figure 3 —figure supplement 1G). We have outlined observed differences between these cell types above and have updated the Results section to clarify that transdifferentiation occurs as a result of both castration and *Pten* loss, albeit resulting in distinct cell states (Lines 427-439).

Line 499: I'm not convinced that these data suggest that specific portions of the intermediate compartment are responsible for their expansion. Without lineage tracing in humans, it's impossible to know whether the change in heterogeneity is the result of increased transdifferentiation of both basal and luminal cells. This may be artificially induced by the promiscuous PB4-Cre promoter rather than a reflection of human prostate cancer as the level of CNVs in basal epithelia in primary tumors appears to be low.

We agree that the source of intermediate heterogeneity is unclear and could originate via multiple mechanisms, including additional phenotype switching events most likely in basal cells or expansion of existing cell states (Lines 415-425, 640-644). In addition, the promiscuity of the Pb-Cre transgenic system could provoke cellular events not found in humans or other systems. Further studies will be necessary to validate them in other contexts and are not within the scope of this manuscript. We have specified appropriate follow-up lineage tracing experiments in the discussion (Lines 643-644).

References

Brady NJ, Bagadion AM, Singh R, Conteduca V, Emmenis LV, Arceci E, Pakula H, Carelli R, Khani F, Bakht M, Sigouros M, Bareja R, Sboner A, Elemento O, Tagawa S, Nanus DM, Loda M, Beltran H, Robinson B, Rickman DS. 2021. Temporal evolution of cellular heterogeneity during the progression to advanced AR-negative prostate cancer. *Nat Commun* 12:3372. doi:10.1038/s41467-021-23780-y

Chang K, Dai B, Kong Y, Qu Y, Wu J, Ye D, Yao X, Zhang S, Zhang H, Zhu Y, Yao W. 2013. Basal cell carcinoma of the prostate: clinicopathologic analysis of three cases and a review of the literature. World J Surg Oncol 11:193. doi:10.1186/1477-7819-11-193

Choi N, Zhang B, Zhang L, Ittmann M, Xin L. 2012. Adult Murine Prostate Basal and Luminal Cells Are Self-Sustained Lineages that Can Both Serve as Targets for Prostate Cancer Initiation. *Cancer Cell* 21:253–265. doi:10.1016/j.ccr.2012.01.005

Crowley L, Cambuli F, Aparicio L, Shibata M, Robinson BD, Xuan S, Li W, Hibshoosh H, Loda M, Rabadan R, Shen MM. 2020. A single-cell atlas of the mouse and human prostate reveals heterogeneity and conservation of epithelial progenitors. *eLife* 9:e59465. doi:10.7554/*eLife*.59465

Dong B, Miao J, Wang Y, Luo W, Ji Z, Lai H, Zhang M, Cheng X, Wang Jinming, Fang Y, Zhu HH, Chua CW, Fan L, Zhu Y, Pan J, Wang Jia, Xue W, Gao W-Q. 2020. Single-cell analysis supports a luminal-neuroendocrine transdifferentiation in human prostate cancer. *Commun Biology* 3:778. doi:10.1038/s42003-020-01476-1

Dong Y, Zhou L, Xia W, Zhao X-Y, Zhang Q, Jian J-M, Gao X, Wang W-P. 2020. Preoperative Prediction of Microvascular Invasion in Hepatocellular Carcinoma: Initial Application of a Radiomic Algorithm Based on Grayscale Ultrasound Images. Frontiers Oncol 10:353. doi:10.3389/fonc.2020.00353

Guo W, Li L, He J, Liu Z, Han M, Li F, Xia X, Zhang X, Zhu Y, Wei Y, Li Y, Aji R, Dai H, Wei H, Li C, Chen Y, Chen L, Gao D. 2020. Single-cell transcriptomics identifies a distinct luminal progenitor cell type in distal prostate invagination tips. *Nat Genet* 52:908–918. doi:10.1038/s41588-020-0642-1

Korsten H, Made AZ der, Ma X, Kwast T van der, Trapman J. 2009. Accumulating Progenitor Cells in the Luminal Epithelial Cell Layer Are Candidate Tumor Initiating Cells in a Pten Knockout Mouse Prostate Cancer Model. *Plos One* 4:e5662. doi:10.1371/journal.pone.0005662

Kwon O, Zhang L, Xin L. 2016. Stem Cell Antigen‐1 Identifies a Distinct Androgen‐Independent Murine Prostatic Luminal Cell Lineage with Bipotent Potential. *Stem Cells* 34:191–202. doi:10.1002/stem.2217

Liu Y, Horn JL, Banda K, Goodman AZ, Lim Y, Jana S, Arora S, Germanos AA, Wen L, Hardin WR, Yang YC, Coleman IM, Tharakan RG, Cai EY, Uo T, Pillai SPS, Corey E, Morrissey C, Chen Y, Carver BS, Plymate SR, Beronja S, Nelson PS, Hsieh AC. 2019. The androgen receptor regulates a druggable translational regulon in advanced prostate cancer. *Sci Transl Med* 11:eaaw4993. doi:10.1126/scitranslmed.aaw4993

Lu T-L, Huang Y-F, You L-R, Chao N-C, Su F-Y, Chang J-L, Chen C-M. 2013. Conditionally Ablated Pten in Prostate Basal Cells Promotes Basal-to-Luminal Differentiation and Causes Invasive Prostate Cancer in Mice. *Am J Pathology* 182:975–991. doi:10.1016/j.ajpath.2012.11.025

Network TCGAR, 2015. The Molecular Taxonomy of Primary Prostate Cancer. *Cell* 163:1011–1025. doi:10.1016/j.cell.2015.10.025

Ninomiya S, Kawahara T, Iwashita H, Iwamoto G, Takamoto D, Mochizuki T, Kuroda S, Takeshima T, Izumi K, Teranishi J, Yumura Y, Miyoshi Y, Asai T, Uemura H. 2018. Prostate Basal Cell Carcinoma: A Case Report. Case Reports Oncol 11:138–142. doi:10.1159/000487389

Sala LS, Boutillon F, Menara G, Goyon‐Pélard AD, Leprévost M, Codzamanian J, Lister N, Pencik J, Clark A, Cagnard N, Bole‐Feysot C, Moriggl R, Risbridger GP, Taylor RA, Kenner L, Guidotti J, Goffin V. 2017. A rare castration‐resistant progenitor cell population is highly enriched in Pten‐null prostate tumours. *J Pathology* 243:51–64. doi:10.1002/path.4924

Shibuya K, Homma S, Yoshida T, Ohno Y, Ichikawa N, Kawamura H, Imamoto T, Matsuno Y, Taketomi A. 2018. Carcinoma in the residual rectum of a long-standing Crohn’s disease patient following subtotal colectomy: A case report. Mol Clin Oncol 9:50–53. doi:10.3892/mco.2018.1626

Song H, Weinstein HNW, Allegakoen P, Wadsworth MH, Xie J, Yang H, Castro EA, Lu KL, Stohr BA, Feng FY, Carroll PR, Wang B, Cooperberg MR, Shalek AK, Huang FW. 2022. Single-cell analysis of human primary prostate cancer reveals the heterogeneity of tumor-associated epithelial cell states. *Nat Commun* 13:141. doi:10.1038/s41467-021-27322-4

Taylor BS, Schultz N, Hieronymus H, Gopalan A, Xiao Y, Carver BS, Arora Vivek K., Kaushik Poorvi, Cerami Ethan, Reva Boris, Antipin Yevgeniy, Mitsiades Nicholas, Landers Thomas, Dolgalev Igor, Major John E., Wilson Manda, Socci Nicholas D., Lash Alex E., Heguy Adriana, Eastham James A., Scher Howard I., Reuter Victor E., Scardino Peter T., Sander Chris, Sawyers Charles L., Gerald WL. 2010. Integrative Genomic Profiling of Human Prostate Cancer. *Cancer Cell* 11–22. doi:10.1016/j.ccr.2010.05.026

Wang ZA, Mitrofanova A, Bergren SK, Abate-Shen C, Cardiff RD, Califano A, Shen MM. 2013. Lineage analysis of basal epithelial cells reveals their unexpected plasticity and supports a cell-of-origin model for prostate cancer heterogeneity. *Nat Cell Biol* 15:274–283. doi:10.1038/ncb2697

Wang S, Garcia AJ, Wu M, Lawson DA, Witte ON, Wu H. 2006. Pten deletion leads to the expansion of a prostatic stem/progenitor cell subpopulation and tumor initiation. *P Natl Acad Sci Usa* 103:1480–1485. doi:10.1073/pnas.0510652103

Xin L, Lawson DA, Witte ON. 2005. The Sca-1 cell surface marker enriches for a prostate-regenerating cell subpopulation that can initiate prostate tumorigenesis. *PNAS* 102:6942–6947. doi:10.1073/pnas.0502320102